# Spatial Transcriptomics-correlated Electron Microscopy maps transcriptional and ultrastructural responses to brain injury

Peter Androvic[1,6], Martina Schifferer [2,3,6], Katrin Perez Anderson[1], Ludovico Cantuti-Castelvetri [2,4], Hanyi Jiang [2,3], Hao Ji[1], Lu Liu [1], Garyfallia Gouna [2,4], Stefan A. Berghoff[2,4], Simon Besson-Girard [1], Johanna Knoferle[2,4,5], Mikael Simons [1,2,3,4] & Ozgun Gokce [1,3,5] ✉

Understanding the complexity of cellular function within a tissue necessitates the combination of multiple phenotypic readouts. Here, we developed a method that links spatially-resolved gene expression of single cells with their ultrastructural morphology by integrating multiplexed error-robust fluorescence in situ hybridization (MERFISH) and large area volume electron microscopy (EM) on adjacent tissue sections. Using this method, we characterized in situ ultrastructural and transcriptional responses of glial cells and infiltrating T-cells after demyelinating brain injury in male mice. We identified a population of lipid-loaded "foamy" microglia located in the center of remyelinating lesion, as well as rare interferon-responsive microglia, oligodendrocytes, and astrocytes that co-localized with T-cells. We validated our findings using immunocytochemistry and lipid staining-coupled single-cell RNA sequencing. Finally, by integrating these datasets, we detected correlations between full-transcriptome gene expression and ultrastructural features of microglia. Our results offer an integrative view of the spatial, ultrastructural, and transcriptional reorganization of single cells after demyelinating brain injury.

Spatial transcriptomics (ST) methods offer spatially-resolved gene expression profiling with single-cell resolution for in-depth characterization of cell types and states within a tissue[1]. While ST is revolutionizing our understanding of tissue organization during homeostasis[2], aging[3], and disease[4], it provides little information on cellular morphology. In contrast, electron microscopy (EM) provides nanometer-resolution view of tissue ultrastructure, but its molecular readouts are limited to techniques like immunogold labeling or correlative light and electron microscopy, which are insufficient for comprehensive characterization of cellular states. Considering the firm connection between cell state and structure[5], integration of molecular and morphological phenotypes has the potential to uncover

functional properties of cells within a tissue. However, techniques that could simultaneously probe cellular ultrastructure and multiplexed molecular profiles in situ are currently unavailable.

Here, we developed spatial transcriptomics-correlated electron microscopy (STcEM). STcEM correlates large-area scanning EM with MERFISH[6] on adjacent tissue sections and links single-cell transcriptional identities with ultrastructural morphologies. Using STcEM, we investigated the regenerative processes in the white matter (WM) of the central nervous system. WM consists primarily of myelinated axons connecting different regions of the brain. Degradation of myelin sheaths and decline of remyelination can result in progressive disease and disability, as seen in chronic progressive multiple sclerosis (MS)[7].

[1]Institute for Stroke and Dementia Research, University Hospital of Munich, LMU Munich, Munich, Germany. [2]German Center for Neurodegenerative Diseases (DZNE), Munich, Germany. [3]Munich Cluster of Systems Neurology (SyNergy), Munich, Germany. [4]Institute of Neuronal Cell Biology, Technical University Munich, Munich, Germany. [5]Department of Neurodegenerative Diseases and Geriatric Psychiatry, University Hospital Bonn, Bonn, Germany. [6]These authors contributed equally: Peter Androvic, Martina Schifferer. ✉e-mail: Ozgun.Goekce@ukbonn.de

To study remyelination, we used mouse model of lysophosphatidylcholine (LPC)-induced WM injury, that is followed by repair process peaking between 14- and 21-days post-injury[8,9]. Successful regeneration depends on the prompt removal of the damaged myelin. Microglia accumulate in demyelinated lesions where they phagocytose and clear damaged, lipid-rich myelin, and initiate a repair process that depends on a precisely coordinated cellular cross-talk[10–12]. This multicellular reaction comprises glial subtypes as well as cells of the adaptive immune system that migrate in small numbers to the lesion where they participate in the injury response[8,13]. In reaction to pathology, microglia undergo structural changes such as deramification, intracellular deposition of phagocytic material, and formation of lipid droplets and inclusions[8]. Structural changes are accompanied by alterations of gene expression[14–17], however, the relationship between diverse transcriptional and morphological states is not well understood.

We used STcEM to characterize spatial gene expression responses to LPC-induced demyelinating injury. We identified small interferon (IFN)-responsive populations of microglia, astrocytes and oligodendrocytes that are associated with injury and occur in proximity of T-cells. By comprehensive analysis of matching EM data, we uncovered ultrastructural heterogeneity of microglia and T-cells and linked it to their transcriptional signatures. This allowed us to identify the morphology of IFN-responsive microglia and transcriptional signature of foamy microglia. We further probed injury-associated pathways of foamy microglia using lipid staining-coupled single-cell RNA sequencing. Finally, we integrated MERFISH, EM and scRNA-Seq datasets to reveal correlations between structural features and transcriptome-wide gene expression in microglia. Thus, we used STcEM to provide an integrated view of the spatial, structural, and molecular organization of demyelinated lesion at single-cell level.

## Results

### Protocol harmonization for correlated ST and EM in a mouse model of demyelination

EM is the gold standard for visualizing structure and organization of cells and myelinated axons due to their size and complexity[18]. MERFISH is a single-molecule spatial transcriptomics technology capable of measuring hundreds to thousands of transcripts simultaneously with subcellular resolution[19]. This makes MERFISH a compelling ST method to integrate with EM, with the shortcoming that sample preparation requirements substantially differ. The standard MERFISH protocol is based on snap-frozen tissue with subsequent washing, embedding and tissue clearing steps that destroy tissue ultrastructure. EM, in contrast, requires high pressure freezing or chemical fixation, heavy metal contrasting and resin embedding steps for imaging, thereby prohibiting subsequent investigation by ST methods. Therefore, we harmonized both protocols (Methods and Tables 1 and 2) until cryomicrotomy allowed adjacent, 10 μm thin coronal sections of the mouse brain to be processed for EM and MERFISH, respectively (Fig. 1A). Using this STcEM method, we collected MERFISH and EM data from a mouse model of LPC-induced demyelinating injury. For comparison, we also generated MERFISH data from a fresh-frozen section prepared with the standard MERFISH protocol.

Ultrastructural analysis requires chemical fixation and cryoprotection by sucrose for a minimum of 25 h before freezing[20]. Despite these long steps, posing potential risk of RNA degradation, MERFISH data obtained with STcEM protocol had similar segmented cell count, cell dropout rate, volume, background signal and showed excellent correlation with the standard protocol, with slightly lower sensitivity (Fig. S1A–C). The MERFISH analysis identified all major brain cell types and localized them to their original spatial positions (Fig. 1B, C). Moreover, by combination of the subclustering analysis and reference mapping, we identified smaller cellular subpopulations such as classes of vascular and immune cells or layer-specific neurons in cortex, whose spatial organization matched expected patterning (Fig. S2A). These

findings were consistent across sections prepared using both the STcEM-modified and standard MERFISH protocol, collectively demonstrating feasibility of STcEM protocol for MERFISH (Fig. S2B, C).

To obtain a comprehensive set of complementary ultrastructural information, we flat embedded the entire coronal cryosection adjacent to the MERFISH section. We sectioned the block face covering both hemispheres ($\sim3 \times 5$ mm$^2$) at 200 nm thickness, which is at the limit of available microtomy tools (Fig. 1D), and serially sectioned a retrimmed ($2 \times 3$ mm$^2$) area bearing the hemisphere with the lesion site. We then used the STcEM data to align ultrastructural changes in cell morphology with cellular transcriptional states in demyelinating lesions. The lesion site was apparent in MERFISH data by profound accumulation of microglia and absence of oligodendrocytes, while the contralateral hemisphere showed none of such characteristics (Fig. 1D). We registered MERFISH data onto EM sections to identify regions of interest where immune and glial cells respond to injury. EM micrographs showed an area with demyelinated axons and cellular debris that overlapped with the lesion area identified by MERFISH (Fig. 1D). Thus, we successfully obtained EM and MERFISH data from adjacent sections, indicating the STcEM protocol's efficacy and potential to connect different spatial measurements.

### Damage-associated responses and spatial organization of glial cells at single-cell resolution

Next, we aimed to characterize transcriptional states and spatial distribution of glial cells responding to demyelinating injury using our MERFISH data. First, we focused on immune cells as the most enriched cluster in the demyelinated lesion. Previous studies revealed the heterogeneity of microglial transcriptional states in different regions and conditions[21], including disease-associated microglia (DAM)[15], activated response microglia (ARM)[17], lipid-droplet-accumulating microglia (LDAM)[22], injury-responsive microglia[16] or IFN response microglia (IRM)[17]. However, spatial as well as annotated ultrastructural information of the different microglial states in and around the lesion is lacking.

The immune cell cluster was composed mostly of microglia (97.0%) identified by microglial signature genes *Selplg*, *Laptm5*, *Csf1r* or *Tmem119*. In addition, we identified a small population (1.6%) of CNS-associated macrophages (CAMs) based on their expression of canonical CAM markers *Mrc1*, *Cd163* and *Msr1*[14]. Notably, we also detected a small population of T-cells (1.4%), marked by *Ptprc*, *Cd3e*, *Cd8a*, *Cd28*, *Cd247* and *Ifng* (Fig. S3A, B). The lesion site was dominated by cells transcriptionally identified as microglia (Fig. S3C), which is consistent with previous fate-mapping study[23]. Subclustering analysis of microglia revealed the presence of four clusters (Fig. 2A, B). The homeostatic cluster, expressing high levels of typical microglial markers such as *Tmem119*, *Csf1r* was enriched in the uninjured hemisphere. The remaining clusters localized predominantly to the injured area indicating they were induced by the demyelinating damage (Fig. 2A–D). The most prevalent cluster was defined by upregulation of DAM markers such as *Cd74*, *Itgax*, *Clec7a* and was enriched in the lesion site, hence we refer to it as DAM-ST. The second cluster shared DAM markers but, in addition, displayed a highly specific upregulation of *Gpnmb*, *Lgals3* and genes related to lipid droplet formation and cholesterol metabolism such as *Plin2*, *Soat1*, *Abca1* and *Abcg1*. This cluster localized to the lesion core area, as opposed to the DAM population which was distributed more equally around the entire demyelinated area (Figs. 2D, 3A). Corresponding areas of the lesion core in the adjacent EM section were filled with lipid droplet-laden foamy microglia (see *Alignment of Ultrastructural Classes with Transcriptional States*), thus we refer to this cluster as "Foamy-ST". The remaining injury-associated cluster was marked by upregulation of interferon-stimulated genes (*Stat1*, *Ifit1*, *Rsad2*, *Usp18*) similar to previously described IFN-response microglia (IRM)[16,17]. To complement the clustering, we investigated gradual changes in microglia states by pseudotime analysis using Monocle3 (Fig. S4). This placed the Foamy-ST

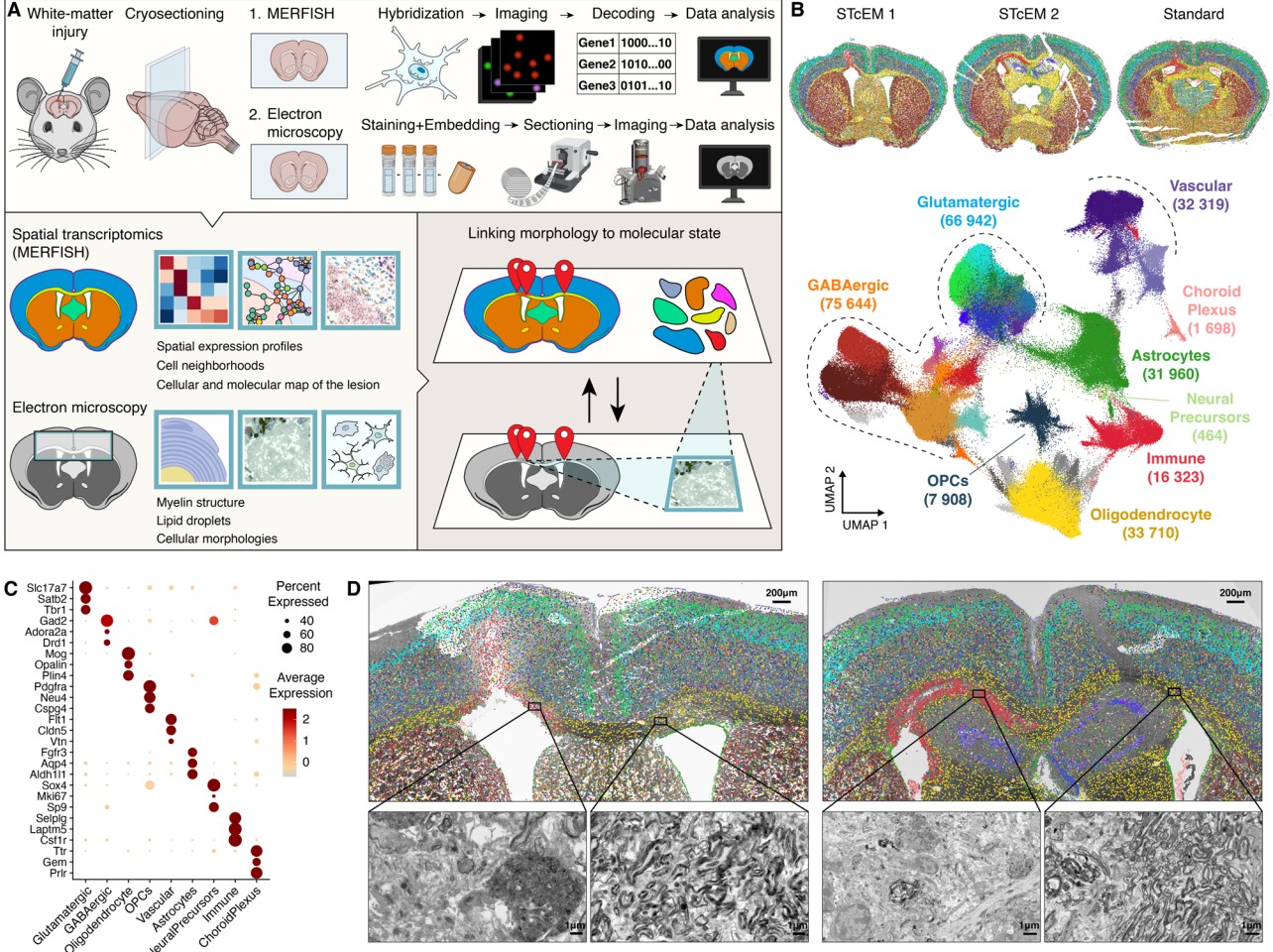

**Fig. 1 | STcEM spatially links single-cell transcriptomes with tissue ultrastructure. A** Overview of the STcEM method. Adjacent sections of the same sample are processed in parallel by harmonized MERFISH and EM protocols, and spatially aligned to link transcriptional profiles with ultrastructure of the regions of interest. **B** Transcriptional identities of single cells with their spatial location in tissue (top) and embedded by UMAP (bottom). "STcEM" refers to modified MERFISH preparation from fixed-frozen sections, "Standard" refers to snap-frozen section. **C** Bubble plot showing expression of cell type markers in identified single cell populations. **D** MERFISH data registered onto the 2D overview EM micrograph. Zoomed-in areas show myelin structure in LPC-injected (left hemisphere) and uninjured (right hemisphere) white matter (corpus callosum). Images from two animals are shown. **A** was created with elements from BioRender.com.

microglia at the end of pseudotime ordering, suggesting they represent the final differentiation stage of the DAM program (Fig. S4B, C). Plotting pseudotime in space showed microglia with highest pseudotime values were located mostly in the center of the lesion (Fig. S4D, E), revealing spatial component to the gene expression transitions of microglia during remyelination.

In addition to microglia, recent studies uncovered transcriptional changes in oligodendrocytes[24,25] and astrocytes[26,27] during neurodegeneration and aging, but their spatial localization with respect to pathology and other cell states remains largely unknown. Thus, we investigated oligodendrocytes as the myelin producing, most abundant white matter cell type that is drastically affected by the LPC-induced injury (Fig. S5). Our analysis identified 5 clusters of oligodendrocytes (excluding oligodendrocyte progenitor cells) (Fig. S5A, B). One cluster represented immature oligodendrocytes marked by *Ctps*, *Dhcr24* and *Tmem163*. Two other clusters represented mature oligodendrocytes: Oligo1 characterized by an upregulation of *Il33*, *Plin4* and *Sgk1*, and Oligo2 by *Klk6*, *Plin3* and *S100b*. Two additional clusters represented injury-responding states of mature oligodendrocytes. One was Oligo3 with elevated *Serpina3n* and *C4b* levels, which have previously been defined as marker genes for disease-associated oligodendrocytes in aging, AD and MS models[24,25,28]. Almost all of the Oligo3 oligodendrocytes were found at the edge of the

demyelinated lesion, while almost no oligodendrocytes survived in the lesion core (Fig. S5C, D). The remaining oligodendrocyte cluster specifically upregulated a battery of interferon-stimulated genes included in our gene panel (*Stat1*, *Ifit1*, *Usp18* and *Rsad2*), similarly to the so-called interferon-responsive oligodendrocytes (IRO) we described in white-matter aging[25]. IRO cluster was overall rare (only 1.7% of all oligodendrocytes), and was found in the lesion as well as in the remaining brain area, although its relative frequency was higher in the lesion edge compared to control white matter (Fig. S5C, D).

We then performed the subclustering analysis for astrocytes and identified five distinct clusters (Fig. S6). Three of these clusters represented states unperturbed by the injury, partially reflecting known regional heterogeneity of astrocytes[29]. Astro1 cluster was marked by *Cspg5*, *Mertk* and *Fgfr3* and localized to both GM and WM regions, while Astro2, characterized by higher expression of *Gfap* and *Aqp4* was enriched in WM regions (Fig. S6A, B). We also found two clusters that were enriched in the lesion area. Among these injury-responding astrocytes were Astro4, which were characterized by the upregulation of *Gfap*, *Serpina3n*, *Vim* and *Fos*, genes previously identified in reactive astrocytes after various pathological insults[26,27]. This cluster localized exclusively to the demyelinated lesion, but in contrast to activated oligodendrocytes populated also the lesion core (Fig. S6C, D). The remaining injury-responding cluster specifically upregulated

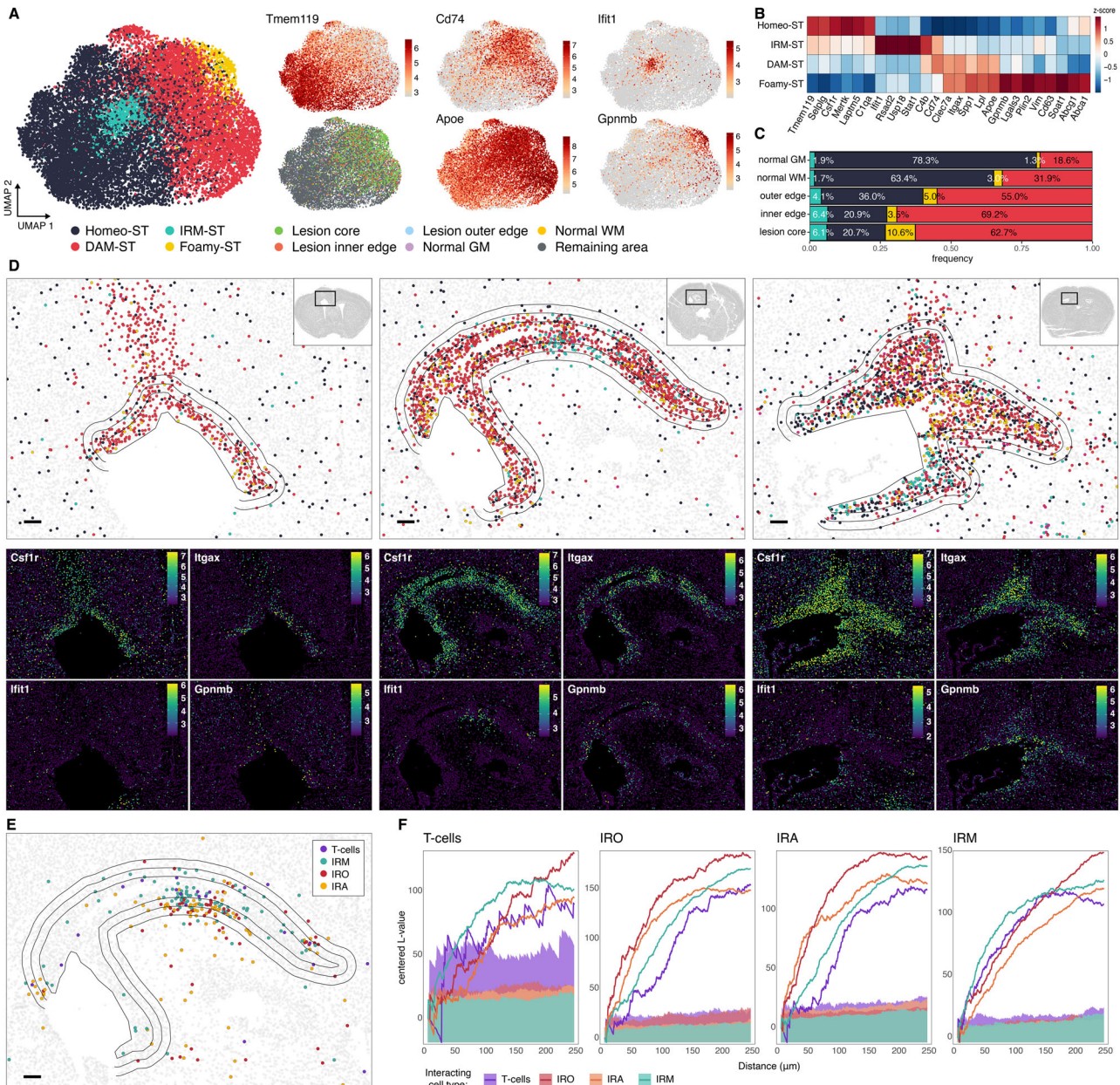

**Fig. 2 | MERFISH analysis of microglia and interferon-responsive cell states.**
**A** UMAP plots of microglia colored by identified clusters (left), tissue region, and expression of marker genes (right). DAM disease-associated microglia, IRM IFN-responsive microglia. **B** Heatmap of average expression per microglia cluster. **C** Frequency of microglia clusters per tissue region. **D** MERFISH spatial plots of microglia clusters in the lesion area of three biological replicate sections (top) and spatial expression of selected marker genes (bottom). Polygons depict segmented lesion areas. Scale bar 100 µm. **E** MERFISH spatial plot of T-cells and IFN-responsive glial states in the lesion area. **F** Centered Ripley's L- functions showing spatial clustering of T-cells and IFN-responsive glial states in the lesion area. Shaded area shows significance envelopes at the 0.05 level (permutation test). IRM interferon-responsive microglia, IRO interferon-responsive oligodendrocytes, IRA interferon-responsive astrocytes.

interferon-stimulated genes, similar to IRM[16,17], thus we refer to it as interferon-responsive astrocytes (IRA). In summary, our data provide a spatial map of cellular states in response to demyelinating injury at single-cell resolution. These states are highlighted by distinct as well as shared gene expression patterns such as altered expression of genes involved in lipid metabolism by microglia, or upregulation of interferon-stimulated genes by all three major glial cell types.

### Interferon-responsive cell states and spatial relationship with T-cells
Previously, IFN-responsive glial states have been separately implicated in CNS pathology[17,25,26,30]. Our finding of a shared IFN signature among

microglia, oligodendrocytes and astrocytes in close proximity suggest a common underlying trigger, such as CD8 T cells, as previously proposed[25]. Thus, we examined the spatial interactions of IFN-responsive glia and T cells. IFN-responsive glia spatially co-clustered in small local niches in the lesion area spanning up to a few hundred microns (Fig. 2E). Moreover, we found that IFN-responsive glia co-localized with T-cells, which were also enriched in the lesion area (Figs. 2E, S7A, B). Quantification of these interactions using Ripley's L function confirmed significant co-clustering (Fig. 2F). The density of T-cells as well as IFN-responsive glia showed considerable animal-to-animal variation. In one MERFISH sample, both T-cells and IFN-responsive glia were absent or nearly absent, respectively, indicating a

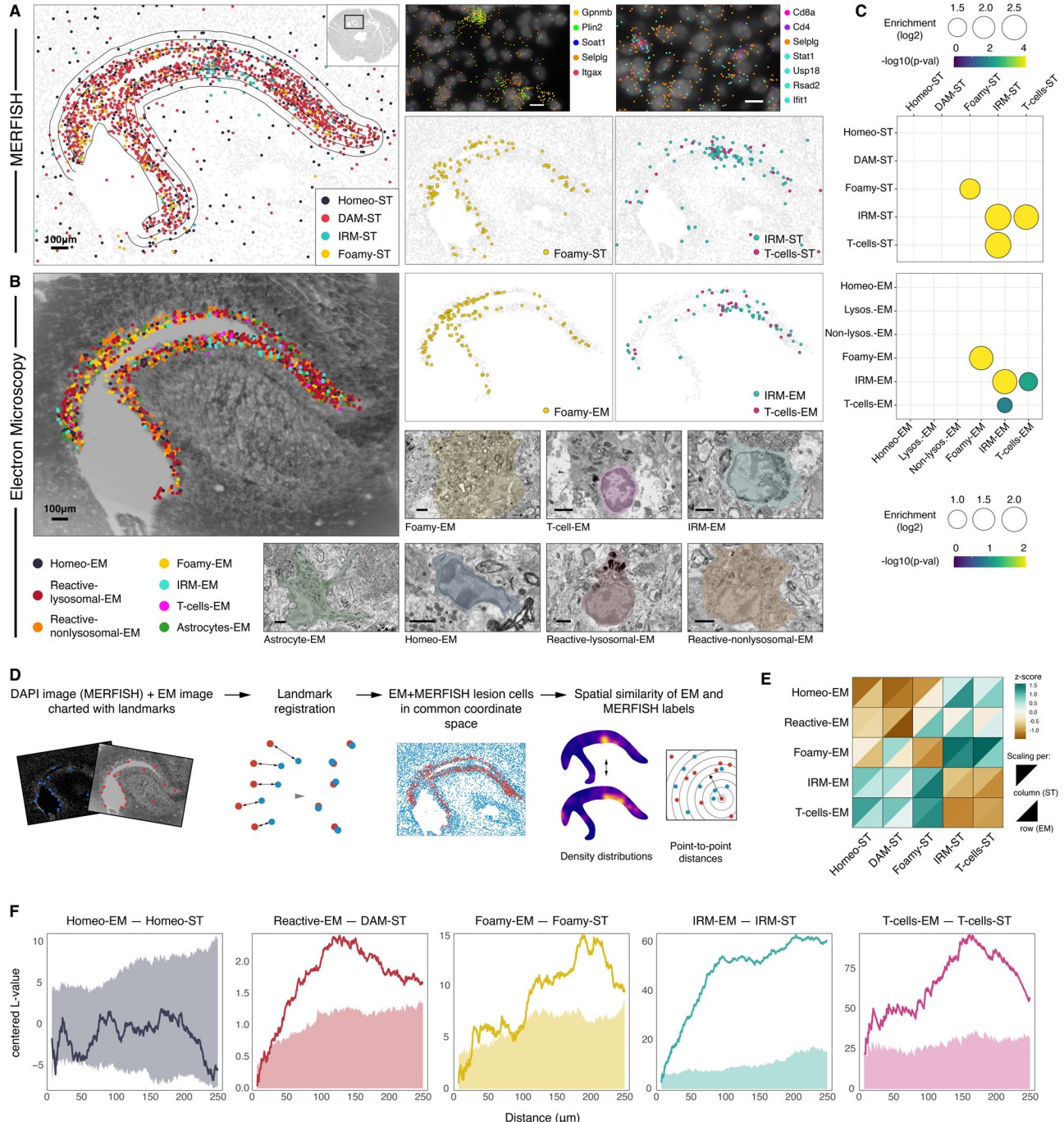

**Fig. 3 | STcEM analysis of microglia and T-cells. A** MERFISH spatial plots of microglia clusters in the lesion area. Zoomed-in plots (top right) show spatial location of individual transcripts of selected marker genes superimposed over DAPI signal. **B** Expert morphological annotation of microglia and T-cells based on EM, and their spatial locations in the lesion superimposed onto a summed EM image stack. Representative EM images per each category are shown, selected from a total of 1059 cells analyzed from one animal. Zoomed-in scale bar 2 μm. **C** Neighborhood analysis of MERFISH data (top) and EM data (bottom) showing enrichment of target cell types (x-axis) among 5 nearest neighbors of query cell type (y-axis). *P*-values and fold-enrichment are derived from empirical distributions obtained from 10,000 random permutations of cell labels. **D** Schematics of the alignment strategy of MERFISH and EM data, and assessment of spatial concordance between identified cell classes. **E** Heatmap of multiscale Earth Mover's distances (EMD) between expert-derived EM classes and ST-derived clusters. Z-scored values per EM class (row scaling) as well as per ST cluster (column scaling) are shown. **F** Centered Ripley's L- functions showing spatial clustering in common coordinate space between expert-derived EM classes and ST-derived clusters. Shaded area shows significance envelopes at the 0.05 level (permutation test).

correlation in the occurrence of these cell populations at the sample level (Fig. S7A, B). Plotting Cd8a vs Cd4 expression to investigate the lineage of T-cells revealed higher Cd8a-positivity, with very few Cd4-positive T-cells detected (Fig. S7C). To confirm the presence of CD8[+] T-cells within the lesion we performed additional staining for CD8[+] T-cells, IBA1[+] microglia, and STAT1[+] cells in another set of mice

(Fig. S7D, E). In all three animals, we found enrichment of CD8[+] T-cells in the lesion compared to the area outside the lesion. Similarly, we found STAT1[+] cells exclusively in the lesion area. In contrast to the MERFISH analysis, almost the entire lesion was filled with STAT1[+] cells in IHC analysis, indicating a difference between STAT1 protein positivity and transcriptional IFN signature. Together, our findings reveal

link between rare T cells and IFN-responsive glial states and its association with demyelinating injury.

## Alignment of ultrastructural classes with transcriptional states

To link the transcriptional and morphological phenotype of injury-responding cells, we collected serial semithin sections of the entire thickness of the adjacent cryotome section. These large area ($2 \times 3$ mm$^2$) sections were used to acquire volume EM data of the lesion area at 20 nm lateral and 400 nm axial (every second section) resolution. While previous correlative light and electron microscopy studies divided tissues into pieces of 1–3 mm edge length[31], we preserved the coronal section throughout processing, which allowed uninterrupted alignment to the MERFISH images. To our knowledge this is one of the largest area scanning EM datasets complementing current large volume contrasting[32] and fast imaging efforts[33].

Blind to MERFISH data, we annotated cells in the demyelinated area into categories according to their ultrastructural morphology[34,35] (Methods and Table 3). Microglia were the most abundant cell type in the lesion center, in agreement with the MERFISH data (Fig. 3A, B). In addition to microglia, we identified a rare population of T-cells and astrocytes in the EM images of the lesion area (Fig. 3B). EM annotation revealed multiple microglial subgroups, including normal-appearing microglia ("Homeo-EM"), reactive microglia with high content of lysosomes ("Reactive-lysosomal-EM"), microglia with reactive morphology, high ER and mitochondrial content, but low lysosomal content ("Reactive-nonlysosomal-EM"), and foamy microglia characterized by excessive deposition of lipid droplets and complex phospholipid inclusions in their cytoplasm[36] ("Foamy-EM"). Foamy-EM microglia displayed similar localization pattern as the Foamy-ST cluster identified by MERFISH in adjacent section, supporting the hypothesis that they represent the same cellular state (Fig. 3A, B right). We also identified another class of microglia, marked by a characteristic heterochromatin pattern and perinuclearly clustered organelles with increased mitochondrial content. Because these microglia co-localized with T-cells and their spatial distribution aligned with positions of IRMs in adjacent MER-FISH section (Fig. 3A, B right), we named them "IRM-EM". Statistical analysis of cellular neighborhoods in MERFISH data revealed that Foamy-ST microglia are significantly clustered together (Fig. 3C top) and confirmed significant co-localization of IRMs and T-cells. Notably, these neighborhood composition patterns were in excellent agreement with neighborhood analysis based on EM data (Fig. 3C bottom), supporting our annotations.

To quantitatively assess the correspondence between EM- and MERFISH-derived cell state labels, we registered the MERFISH and EM lesion data into a common coordinate space (Methods) and subsequently evaluated the similarity of spatial patterns between EM and ST classes (Fig. 3D-F). As a first metric, we used multiscale Earth Mover's distance (EMD) to compare the similarity of 2D density distributions of each cell class (Methods) (Fig. 3E). We found that foamy-ST microglia bore the closest resemblance to foamy-EM microglia, and vice versa, as evidenced by their low EMDs. Similarly, IRMs and T-cells were relatively closest to their proposed counterparts in the other modality as well as to each other. Patterns of reactive (merged non-lysosomal and lysosomal) and homeostatic microglia populations were also similar across modalities, although this analysis could not unambiguously differentiate between the two cell states due to their similar patterning in the lesion area. As a second metric, we used Ripley's L function to quantify co-clustering between pairs of cells, reasoning that matching cell labels between modalities would be found in proximity more frequently compared to random labeling (Fig. 3F). Our analysis confirmed this for all matching pairs except homeostatic microglia. Overall, the quantitative assessments support the qualitative observations. Together, our results demonstrate how the integration of morphological and molecular phenotypes by STcEM can assign identity to unknown cell states, and link ultrastructural morphology with their transcriptional profiles.

## Feature-based quantitative analysis of ultrastructural heterogeneity

After identifying transcriptional and morphological cell states and their spatial organization, we wanted to understand the underlying structural heterogeneity in more detail. To this end, we segmented eight distinct subcellular structures and the respective cell boundary in EM images of microglia and T-cells located in the lesion (Fig. 4A and Methods). For each of the 933 cells, we selected the plane with the largest cellular area for two-dimensional segmentation, resulting in a final set of 32 structural metrics after calculating the area of encompassing structures and ratios (Fig. 4A and Methods). Representative EM images of cells and their segmented intracellular structures are shown in a zoom-in view in Fig. 4B. To obtain spatial resolution, we recorded the global positions of each cell in the stitched and z-projected overview image (Fig. 4D).

Unsupervised clustering of this data identified six ultrastructurally distinct cell clusters (Fig. 4B–D). Cluster 5 showed a foam cell morphology, characterized by a large cell area and high amount of lipid droplets. Cluster 4 was marked by a high ratio of nuclear to cytoplasmic area and very low amount of all cytoplasmic organelles, reminiscent of the T-cell morphology[37]. Cluster 3 was characterized by high ratios of mitochondria to other organelles, a higher ratio of empty cytoplasm to organelle-filled cytoplasm and its location largely corresponded to the site where we identified IRMs, suggesting this cluster represents IRMs. Cluster 2 was characterized by high lysosomal content, while Clusters 1 and 0 were less distinctive, and seemed to represent microglia in a more homeostatic and Reactive-nonlysosomal state, respectively.

To evaluate the correspondence between expert-assigned cell labels and structural clusters, we calculated overlap of these categories (Fig. 4E). We found significant overlaps with most labels matching one-to-one, showing agreement between the results of the two analyses. In addition, we compared the spatial distributions of structural and MERFISH-derived clusters using EMD in common coordinate space (Fig. 4F). This confirmed similarity between cluster 5 and Foamy-ST, Cluster 3 and IRM-ST, and Cluster 4 and T-cells-ST. In summary, our feature-based analysis supports expert-assigned cell categories and reveals quantitative structural characteristics of microglia and T-cells responding to demyelination.

## Comprehensive characterization and validation of foamy microglia

We aimed to perform a deeper characterization and validation of microglial populations identified by MERFISH, using full-transcriptome analysis. In particular, our STcEM analysis indicated that Foamy-ST cluster identified by MERFISH represents lipid-loaded microglia with foamy morphology. Foamy microglia are key players in lipid metabolism following demyelinating injury, but their molecular identity and underpinnings of their emergence and fate remain poorly understood[36,38,39]. With this in mind, we performed single-cell RNA sequencing (scRNA-Seq) on sorted CD11b$^+$ cells in a new cohort of mice using our established, artifact-free SmartSeq2 protocol (Fig. S8A)[40,41]. We also stained the cells with BODIPY, a fluorescent dye binding neutral lipids that is often used to detect cellular lipid droplets (Fig. S8B). We then employed an index-sorting strategy, recording the BODIPY fluorescence signal for each sorted cell. This allowed us to assign quantitative BODIPY values to single cell transcriptomes during bioinformatic analysis, as a measure of the cell's lipid content.

After QC and exclusion of smaller contaminating cell populations, we obtained 1017 single microglial transcriptomes from 3 control and 5 LPC-injected mice (Fig. S8C, D). We then mapped MERFISH-derived cluster labels onto scRNA-Seq data, which revealed

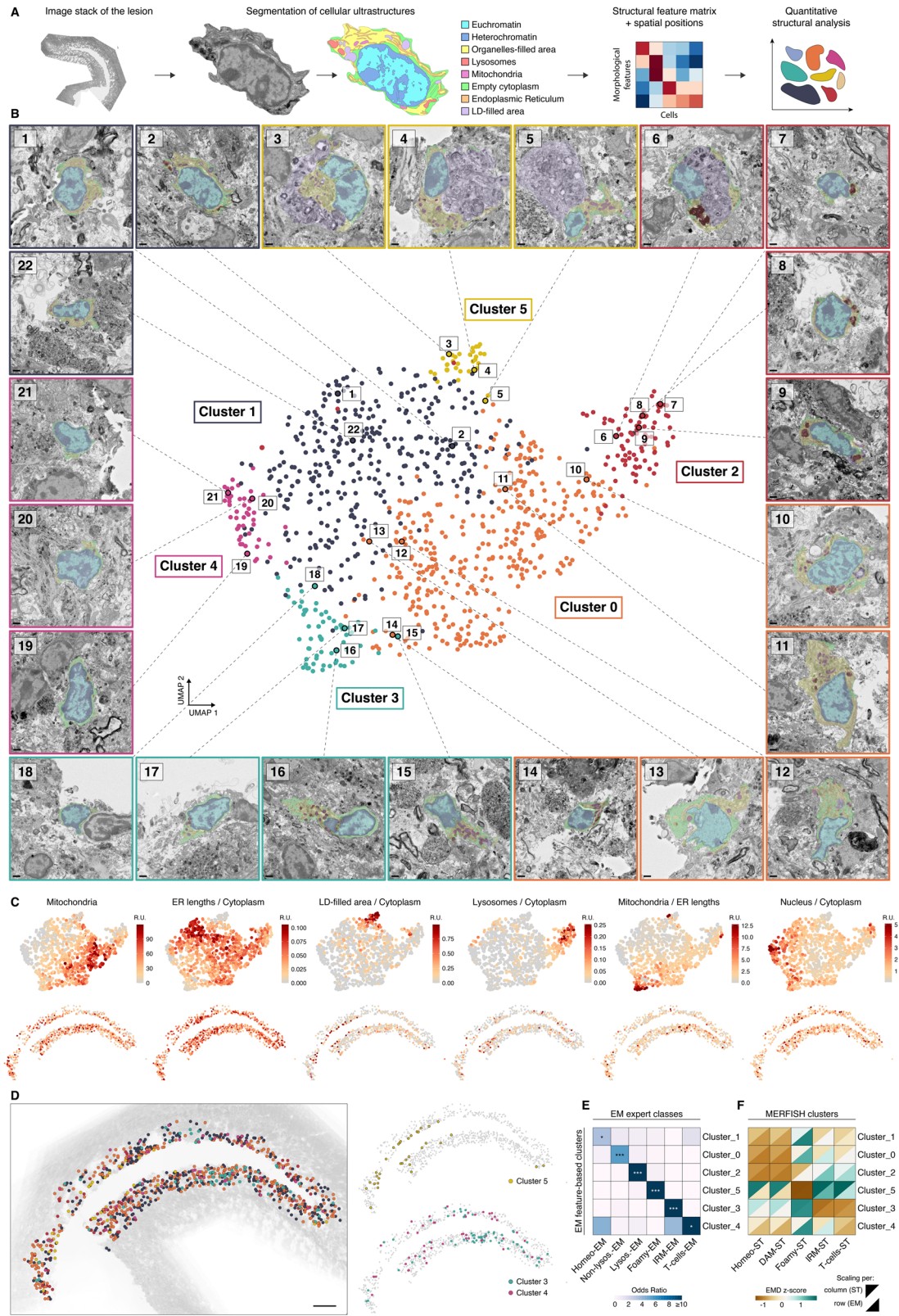

**Fig. 4 | Segmented feature-based structural analysis. A** Overview of the analysis pipeline. Subcellular structures of microglia and T-cells were segmented in the EM images of the lesion area, quantified, and analyzed by unsupervised clustering. **B** UMAP embedding of single-cell morphologies, colored by identified clusters. Representative EM images from each cluster are shown, selected from a total of 933 cells analyzed from one animal. Scale bar 1 μm. **C** UMAP plots (top) and spatial plots (bottom) colored by structural feature values. **D** Spatial plots of structural clusters superimposed onto the EM image stack of the lesion. **E** Heatmap of overlaps between expert-derived EM classes and feature-based structural clusters. One-sided Fisher exact test. Asterisks show nominal significance (*p-val < 0.1, ***p-val < 1e⁻⁵). **F** Heatmap of multiscale Earth Mover's distances (EMD) between feature-based structural clusters and ST-derived clusters. Z-scored values per structural cluster (row scaling) as well as per ST cluster (column scaling) are shown.

the presence of equivalent microglial populations with highly concordant transcriptional profiles (Fig. 5A, B). Leveraging the full-transcriptome data, we performed functional enrichment analysis focusing on the foamy microglia cluster, which revealed a strong upregulation of pathways related to lipid metabolism and lysosomal processing, showing that these cells are engaged in lipid-processing functions (Fig. 5C). Analysis of gene expression scores of published single-cell signatures of lipid-associated macrophages from various tissues[42–44] revealed that our foamy microglia cluster upregulated consensus lipid-associated macrophage signature, while the remaining clusters upregulated either homeostatic, interferon or DAM signatures (Fig. 5D). Single-cell analysis of BODIPY staining data showed that foamy microglia cluster had on average highest BODIPY values among all four clusters (Fig. 5E). Overall, scRNA-seq analyses support STcEM annotations and indicate conserved,

lipid metabolism-related gene expression responses in foamy microglia and peripheral lipid-loaded macrophages.

Our dataset provides an opportunity to examine correlations between gene expression and lipid deposition in microglia by measuring BODIPY values for individual RNA-sequenced cells. To evaluate these dependencies, we adapted the scWGCNA workflow[45] for single-cell correlation analysis. This approach groups transcriptionally similar cells into metacells to robustly calculate correlations by accounting for sparsity and noise known to hamper single-cell correlation analyses[46,47] (Methods). We found that top genes positively correlating with BODIPY were largely foamy microglia signature genes such as *Gpnmb*, *Lgals3, Apoe, Capg, Apoc1, Anxa5, Fabp5, Ftl1, Fth1, Ctsb*, among others, while genes associated with homeostatic microglia such as *Tmem119*, *Selplg*, *Cst3* showed negative correlation (Fig. 5F). Plotting gene expression vs BODIPY in metacells showed that Foamy-ST markers *Gpnmb, Apoc1* and *Lgals3*, among others, have delayed increase followed by a sharp rise, reaching highest levels in metacells with highest BODIPY values, consistent with their classification as markers of lipid-loaded microglia (Fig. 5G). Gene set enrichment analysis confirmed that genes increasing with BODIPY are enriched in functional terms related to cholesterol homeostasis, lipid storage, cholesterol and lipid transport, or regulation of lipid metabolism. In addition, we found enrichment of genes associated with lysosomes, oxidative phosphorylation and respiratory chain, antigen processing and presentation, ribosomal proteins, and cell death pathways (Fig. 5H).

To further validate the link between foam cell morphology and molecular signature, we co-stained IBA1+ microglia with antibodies against intracellular lipid droplet protein PLIN2 and Galectin3 (encoded by Foamy microglia signature gene *Lgals3*), in sections obtained from a new set of LPC-injected mice (n = 5) (Fig. 5I). Consistent with our previous data, Galectin3 labeled 90.8% of PLIN2+ IBA1+ microglia, but only 18.7% of PLIN2- IBA1+ microglia, representing significant, nearly 5-fold enrichment in lipid droplet-containing microglia (Fig. 5J, K). In addition, we observed co-labeling of PLIN2+ IBA1+ microglia with antibodies against Glycoprotein NMB, encoded by *Gpnmb*, the top marker gene of Foamy microglia (Fig. 5I). Altogether, lipid staining-coupled scRNA-Seq and immunocytochemistry experiments validated and expanded the findings from the STcEM analysis by identifying transcriptome-wide profiles of microglial clusters and revealing correlations between lipid deposition and gene expression in microglia.

## STcEM enables gene-structure correlation analysis

So far, we have used STcEM to identify and match morphological and transcriptional cellular states. We sought to take the STcEM method a step further and examine its ability to recover continuous dependencies between ultrastructural features and gene expression (Fig. 6). Such gene-structure correlation networks could reveal functional modules and identify relationships between morphological properties and gene expression beyond cell state categories. To obtain genome-wide coverage, we imputed expression of unmeasured genes in

MERFISH microglia data from our SmartSeq2 dataset (Methods), leading to 287 measured and 8431 imputed genes. Comparison of imputed gene expression to ground truth showed consistent patterns across microglial clusters (Fig. S9). We then devised a strategy for spatial transfer of the gene expression and morphological feature values between neighboring cells and across the modalities (Fig. 6A). Specifically, for 933 EM and 986 neighboring MERFISH cells in the lesion area, we calculated distance-weighted averages of the gene expression, and structural features for each cell based on its neighbors in common coordinate space. This resulted in smoothed values of both modalities in all 1919 cells (Methods). We then calculated Spearman correlation between genes and structural features.

We found that values of several structural features are correlated with functionally-related genes (Fig. 6B). For instance, endoplasmic reticulum (ER)-related gene ontology cellular component (GOCC) terms were significantly overrepresented among top genes positively correlating with ER area (e.g. "ER membrane"). Similarly, mitochondria-related terms (e.g. "mitochondrial matrix") were overrepresented among genes positively correlating with mitochondria area. Other examples include overrepresentation of heterochromatin-associated terms ("rDNA heterochromatin") and protein complexes active in pericentric heterochromatin ("Sin3 complex") and nucleus ("ESC/E(Z) complex") among genes positively correlating with heterochromatin area; as well as overrepresentation of cytoplasm-associated GOCC terms among genes negatively correlating with heterochromatin area (e.g. "septin cytoskeleton", "microtubule"). These results demonstrate the ability of STcEM to capture biologically meaningful associations between structure and gene expression, even though the EM and MERFISH data were generated from adjacent sections.

We further focused on associations between genes and morphological features related to microglial reactive states, visualizing them as heatmaps (Fig. 6C, D). In addition to visualizing the correlation structure of the features, we show their average abundances in the identified morphological and transcriptional clusters, as well as spatial matching of the clusters (Fig. 6C, D). This revealed that spatially-transferred and directly measured values are largely in agreement for matching structural and transcriptional clusters. We were able to show that interferon-stimulated genes such as *Stat1, Ifit1, Rsad2, Irf7, Mx1*, that define IRM microglial state, positively correlated with mitochondrial content in a cell, ratio of mitochondria to other organelles and values of empty cytoplasm area, which are the structural features also marking morphological cluster 3. Further, we found that MHCII genes such as *Cd74, H2-Ab1, H2-Aa, H2-Oa*, that are most upregulated in DAM cluster, were positively correlated with cellular ER content. MHCII genes also showed tendency to increase with ratio of organelles-filled cytoplasm to lipid droplet-filled cytoplasm, suggesting potential differences in antigen-presenting capacity between DAM and Foamy microglia clusters. Although the correlations between Foamy microglia signature genes and lipid droplet area were more elusive in this analysis, we did detect positive dependencies, including *Gpnmb* and *Apoe*. Furthermore, foamy signature genes showed positive correlation with lysosomal content, which is consistent with our observation of microglia containing lipid-rich cytoplasm interspersed with lysosomes (see cell number 6 in Fig. 4B). In summary, here we used STcEM to begin to unravel dependencies between gene expression and ultrastructural features pertaining to microglial reactive states, providing a more systematic perspective on the relationships between microglial structure and molecular function.

## Discussion

ST technologies have opened new horizons in biology; however, ST measurements are typically disconnected from one of the most fundamental readouts of tissue phenotype—cellular morphology. While some studies have linked paired histological staining with lower-resolution ST methods[48,49], these approaches are unable to detect

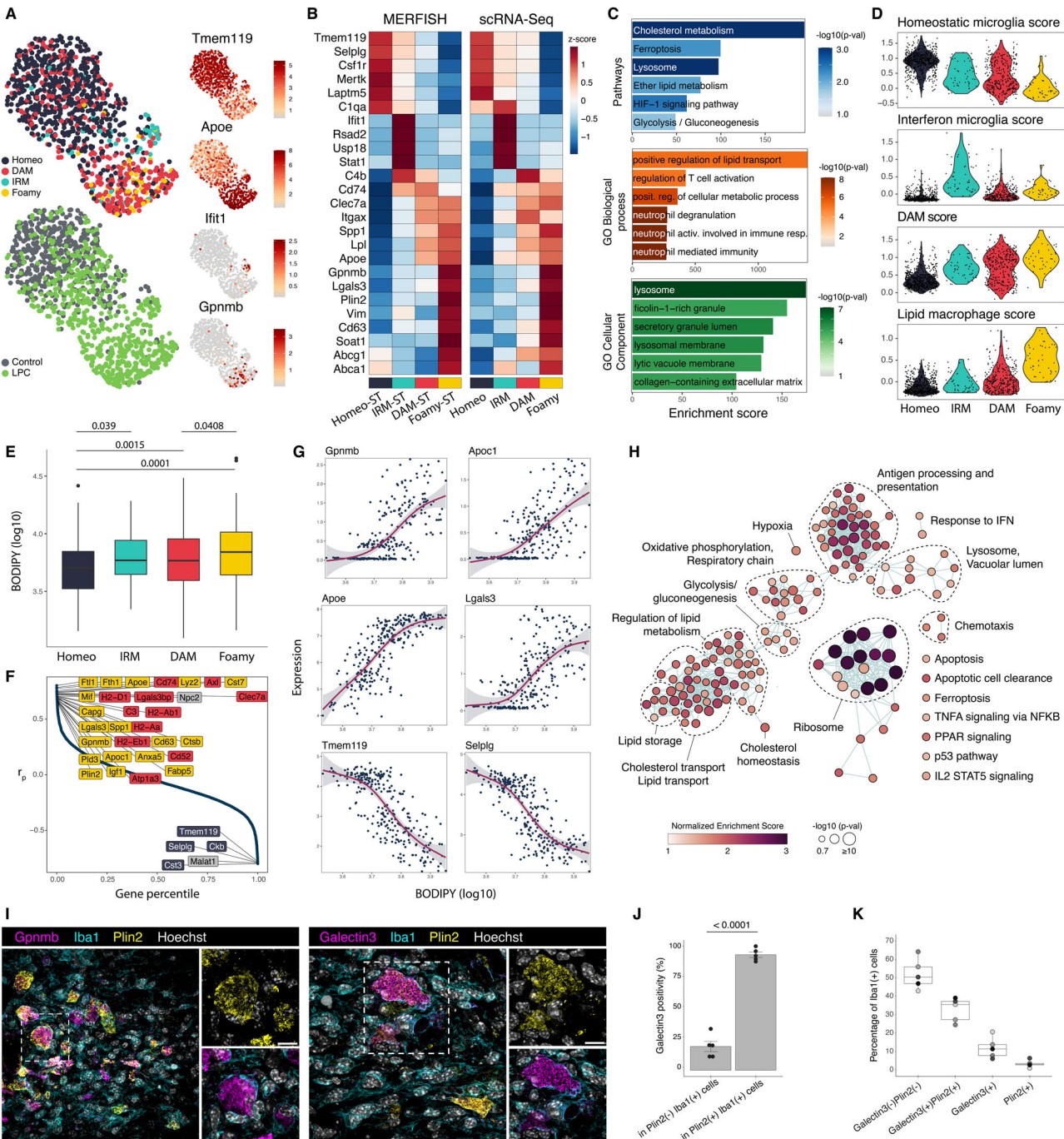

**Fig. 5 | Extended characterization and validation of microglial populations.**
**A** UMAP embedding of microglial scRNA-Seq data colored by cluster (top left), experimental group (bottom left) and expression of selected markers (right).
**B** Heatmap of average expression per microglial cluster measured by MERFISH (left) or by scRNA-Seq (right). **C** Pathway and Gene Ontology enrichment analysis of foamy microglia signature genes. One-sided Fisher exact test, nominal *p*-values.
**D** Violin plots showing single-cell activity scores of homeostatic microglia, interferon-stimulated microglia, disease-associated microglia, and lipid-associated macrophages expression signatures collected from literature. **E** Boxplot of BODIPY fluorescence values of single microglia cells index-sorted from LPC-injected mice. Nominal *p*-values of post-hoc pairwise comparisons (two-tailed) following ANOVA, *n* = 560 cells obtained from 5 animals. Boxplots display the median (central line), interquartile range (IQR; box), and 1.5 * IQR (whiskers). Points beyond the whiskers represent outliers. **F** Rank plot of Pearson correlation coefficients between gene

expression and BODIPY values in metacells. **G** Scatter plots of metacell gene expression vs BODIPY values for selected genes. Trendline represents generalized additive model fit, error band displays 95% confidence interval. **H** Network of pathways and Gene Ontology terms enriched in genes positively correlating with BODIPY. Gene Set Enrichment Analysis test, nominal *p*-values are shown. **I** Stainings of GPNMB, Galectin3 (Foamy microglia signature genes), IBA1 (microglia) and PLIN2 (lipid droplets) in LPC-injected white matter. Scale bar 10 μm. **J** Quantification of Galectin3 positivity in PLIN2-positive and PLIN2-negative microglia. Paired *T*-test (two-tailed, *p*-val = 4.021e$^{-6}$), *n* = 5 animals. Data are presented as mean values +/- SEM. **K** Quantifications of Galectin3 and PLIN2 single and double positive Iba1+ microglia. *N* = 5 animals. Boxplots display the median (central line), interquartile range (IQR; box), and 1.5 * IQR (whiskers). Source data are provided as a Source Data file.

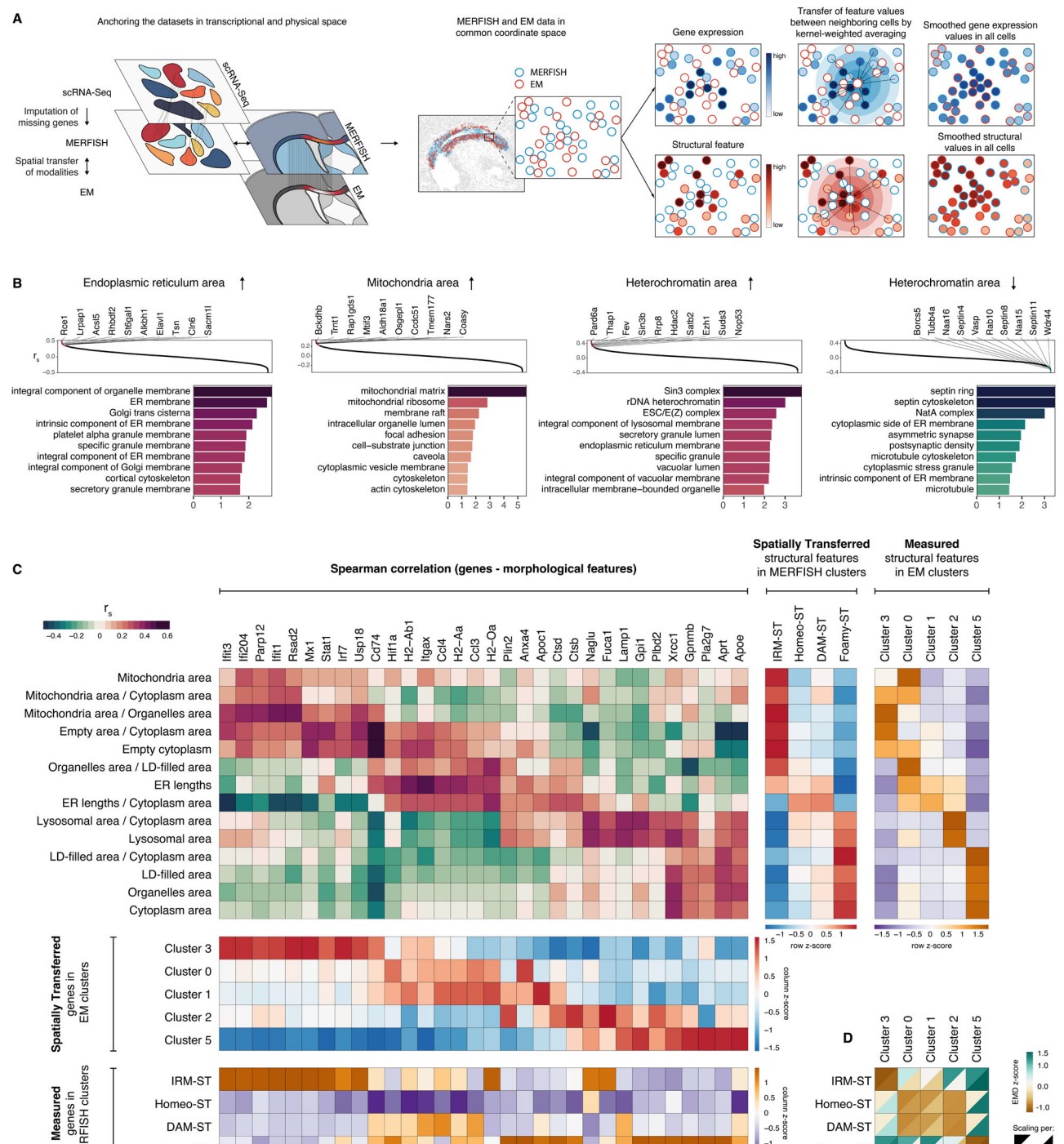

**Fig. 6 | Correlation of gene expression and structural features in microglia.**
**A** Schematics of the computational strategy for spatial transfer of modalities followed by gene-structure correlation analysis. Unmeasured genes in MERFISH are imputed from SmartSeq2 data. MERFISH and EM data from adjacent sections are registered to common coordinate space. Gene expression values as well as structural feature values are then transferred between neighboring cells using distance-weighted averaging. **B** Rank plots of Spearman correlation between genes and selected structural features (top). Barplots on the bottom show enriched Gene Ontology Cellular Component terms among the top 200 positively (shades of red) or negatively (shades of green) correlating genes with respective structural feature. One-sided hypergeometric test, nominal *p*-value. **C** Heatmaps showing correlation between selected genes and structural features in microglia (red-green palette), average values of spatially transferred modality in ST and EM clusters (red-blue palette), and average values of directly measured modality in ST and EM clusters (orange-purple palette). **D** Heatmap of multiscale Earth Mover's distances (EMD) between feature-based structural clusters and ST-derived clusters. Z-scored values per structural cluster (row scaling) as well as per ST cluster (column scaling) are shown. **A** was created with elements from BioRender.com.

single cells or their subcellular organization. Attempts to link ultrastructure to gene expression have been scarce, and limited to invertebrates with stereotypical cellular anatomy[50]. In this study, we integrated two seemingly incompatible high-resolution methods - EM and MERFISH - by anchoring them in spatial dimension. This allowed us

to map and link ultrastructural and transcriptional phenotypes of single cells responding to LPC-induced demyelination.

After LPC injury, microglia accumulate at the injury site to clear cellular debris and initiate the regenerative process[51]. Phagocytosis of lipid-rich myelin drives the formation of foamy microglia, which play

key roles in pathophysiology of multiple sclerosis[51], but also in aging[52] and neurodegenerative conditions such as Alzheimer's disease (AD)[53]. Using STcEM, we uncovered their localization and transcriptional profile, which was characterized by the specific upregulation of *Gpnmb*, lysosomal transcripts, and genes involved in cholesterol and lipid metabolism. Intriguingly, in patients with early active MS, a microglia cluster with highly similar signature has been identified as the most strongly associated with the disease[54]. A similar gene set, including *Gpnmb*, is upregulated in lipid-loaded macrophages in atherosclerotic lesions[43], obesity[42] or fatty liver disease[44], suggesting a conserved response to lipid overload in microglia and macrophages across tissues and conditions. GPNMB represents a genetic risk factor for Parkinson's disease (PD)[55], and is elevated in the substantia nigra of PD patients[56], and in spinal cords of amyotrophic lateral sclerosis (ALS) patients[57,58]. Our results concur with the function of *Gpnmb* within a lipid-associated inflammatory network and identify it as a marker of subpopulation of foamy microglia. In addition, our data show that foamy microglia upregulate several AD genetic risk factors, including cholesterol transporters *Abca1*, *Apoc1* and *Apoe*[59], further implicating them in the disease beyond the LPC demyelination model. STcEM thus identified candidate genes that could facilitate therapeutic targeting of foamy microglia to promote regeneration and decrease neuroinflammation in demyelination disorders, aging and neurodegeneration.

Following LPC injury, an early and transient influx of T cells plays a role in microglia activation and subsequent demyelination[13]. In this study, we observed low T-cell numbers with considerable animal-to-animal variation, possibly due to the late analysis time point when T cells had already entered and were lost. However, we found that the few remaining T cells were enriched in the lesion, and co-localized with IFN-responsive cell states. This suggests that T cells, similar to aging WM[25], could contribute to induction of IFN-responsive cell states and play a role in tissue damage. STcEM allowed us to discover structural properties of IFN-responsive microglia, characterized by areas of empty cytoplasm and perinuclearly clustered organelles with increased proportion of mitochondria. Interestingly, a recent study reported that transcriptional response to IFNγ in macrophages is dependent on mitochondrial function and mitochondrial mass[60], but more research will be needed to elucidate the impact of IFN signaling on the ultrastructural organization of cells. In contrast to previous EM analysis of microglia in AD models[61], we did not observe a "dark" microglia phenotype, indicating potential differences in microglial remodeling during chronic pathologies and acute demyelination.

The recent surge in generating vast EM datasets also increased the need for their annotation and analysis[62,63]. However, identifying cell types from EM images is time-consuming and requires expertize. STcEM tackles this bottleneck and extends the capacity of current EM methods such as correlative light and electron microscopy or immunogold labeling from handful of antibodies to hundreds of molecular targets. This enables the identification of subtle cellular states, and the study of interplay between structural reorganization and gene activity of cells in response to stimuli in situ. This is showcased by our gene-structure correlation analysis, that detected low to moderate correlations, which aligns with other studies[64,65], and points to the complementary nature of gene expression and structural morphology in defining multidimensional cellular phenotypes. Our vision for future work is to utilize reference STcEM datasets from the same tissue to train models for prediction of single-cell gene expression from EM images or vice versa[48].

While in this study we focused on the brain, STcEM can, in principle, be applied to any tissue, opening opportunities across fields. For example, STcEM could be applied to simultaneously identify virus particles via EM-based morphological criteria[66] and detect associated spatial gene expression responses. Our approach to anchoring modalities via spatial domain could be expanded to other spatial technologies that are otherwise incompatible, facilitating spatial multiomics. We envision future STcEM developments, where multiple sections from the same cell could be generated by ultrathin sectioning, eventually providing a powerful approach for subcellular omics.

## Methods

### Animals
All mouse experiments in this study were performed with the approval and according to the regulations of the District Government of Upper Bavaria and reported according to guidelines[23,24]. Male C57BL/6 J mice were obtained from Janvier Laboratories (Supplementary Data 1). All mice were housed at the animal facility in the German Centre for Neurodegenerative Diseases (DZNE) in Munich in standard, pathogen-free conditions. The temperature in the housing unit was kept between 20 and 22 °C with 40–60% humidity and a 12-h light/12-h dark cycle.

### LPC injections
Lysophosphatidylcholine (LPC) injections were administered at the age of 3–4 months. A solution of 1% LPC (L4129, Sigma) in PBS was mixed with Monastral blue (274011, SigmaAldrich) at a concentration of 0.05% to aid with visualization of the lesion during tissue processing. Mice were anaesthetized with an intraperitoneal injection of MMF solution (0.5 mg medetomidin/kg (body weight), 5.0 mg midazolam/kg (body weight) and 0.05 mg fentanyl/kg (body weight)). Then, head fur was removed, the eyes were treated with bepanthene cream (1578847, Bayer) and a small incision in the skin was performed to expose the skull. The mouse was positioned into a stereotactic injection apparatus and a small hole was drilled at the following injection coordinates (from bregma): (X, ± 1.0 mm; Y, −0.1 mm). A glass capillary containing the LPC–monastral blue solution was then lowered to Z: −1.40 mm (including scull) from bregma, and 1 μL was injected at a rate of 100 nL/min. Two minutes after the delivery of LPC, the capillary was slowly retracted. The mouse was then injected with 0.05 mg buprenorphin/kg (body weight), and the skin was sutured. Anesthesia was terminated by a subcutaneous injection of AFN solution, containing 2.5 mg/kg (body weight) atipamezol, 1.2 mg/kg (body weight) naloxon and 0.5 mg/kg (body weight) flumazenil.

### Tissue collection and preparation for STcEM
Eighteen days after LPC injection, mice were anaesthetized with an i.p. injection of MMF (fentanyl (0.05 mg/kg)–midazolam (5 mg/kg)–medetomidine (1 mg/kg)) and transcardially perfused with 2 UI/mL Heparin (Heparin-Natrium-25000-ratiopharm®, PZN: 03029843) in HBSS (no calcium, no magnesium, Gibco™, 14175129) for 3 min and 4% paraformaldehyde (PFA, EM Grade, Electron Microscopy Sciences, Cat. No. 15710, diluted in 10X PBS, Invitrogen™ (to a final concentration 1x), AM9624 and UltraPure™ Distilled Water, Invitrogen™, 10977-035) for 5 min, before carefully removing the brain from the skull. Afterwards, the brains were fixed by submerging in 4% PFA for 6 h, followed by 14 h in 15% sucrose (Sigma-Aldrich, S0389 in UltraPure™ Distilled Water, Invitrogen™, 10977-035) and 5 h in 30% sucrose. Next, the PFA-fixed brains were simultaneously embedded in Tissue-Tek® O.C.T.™ Compound (Sakura, 4583) and frozen in a plastic mold on dry ice. For the "standard protocol", fresh frozen brain samples were used: the mouse was only perfused with Heparin in HBSS and directly embedded and frozen in Tissue-Tek® O.C.T.™ Compound on dry ice. Brains were stored at −80 °C until further processing.

### Tissue sectioning
Coronal, 10 μm thick brain sections were prepared and collected at a cryotome (CryoStar NX70, Thermo Scientific). Sections determined for MERFISH analysis were placed on round glass slides provided by Vizgen Corp. (Cambridge, MA 02138; MERSCOPE slide part number 20400001). Sections for electron microscopy were collected on Superfrost Plus® Gold slides (Thermo Scientific, Menzel Gläser, K5800AMNZ72). Brain sections for MERFISH were subsequently washed two times with PBS and one time with 70% ethanol (VWR

**Table 1 | Comparison of protocol changes between standard MERFISH and STcEM MERFISH sample preparation**

| | Standard MERFISH protocol | StcEM MERFISH protocol | reasoning | impact |
|---|---|---|---|---|
| **Tissue perfusion** | Perfusion with HBSS (Ca2+, Mg2+ free) | Perfusion with HBSS (Ca2+, Mg2+ free), followed by perfusion with 4% PFA (EM grade quality) in PBS-CMF | Fixation needed for EM protocol | Longer perfusion time poses potential risk of losing mRNA and induction of stress-response genes |
| **Tissue preparation and freezing** | Freezing in Tissue-Tek® O.C.T.™ Compound on dry ice. Storage at –80 °C. | Incubation at 4 °C for 6 h in 4% EM grade PFA in PBS-CMF, followed by 14 h in 15% sucrose and 5 h 30% sucrose in PBS-CMF, before embedding and freezing in Tissue-Tek® O.C.T.™ Compound on dry ice. Storage at –80 °C | Fixation needed for EM protocol | Potential risk of nuclease contamination and lowering RNA quality and quantity |
| **Tissue Sectioning** | cryotome sections, 10 µm thickness | cryotome sections, 10 µm thickness | | |
| **Post-preparation and post- fixation** | Washed two times with PBS, followed by one time 70% ethanol for 5 min each | Washed two times with PBS, followed by one time 70% ethanol for 5 min each | | |
| **Storage** | 70% ethanol | 70% ethanol | | |

Chemicals, 20.821.310, diluted in UltraPure™ Distilled Water, Invitrogen™) for 5 min each. Then samples were individually sealed in bags filled with 70% ethanol and shipped to Vizgen Corp. (Cambridge, MA 02138) for MERFISH analysis. Sections for electron microscopy were stored at −80 °C until further processing.

## MERFISH procedure

**Gene panel.** Gene panel for this study consisted of 287 protein-coding genes and 98 blank probes. Genes included selection of known brain and immune cell type markers including glial cells, vascular cells, T-cells, macrophages and subtypes of Glutamatergic and GABAergic neurons. In addition, we included a panel of microglial, astrocytic and oligodendrocytic reactive markers from literature, and genes from cholesterol metabolic pathways. Full gene panel is listed in Supplementary Data 2.

**Hybridization.** Samples on beaded slides were placed tissue-side up in 60×15 mm petri dishes and kept at the back of the cryostat at −20 °C for at least 5 min for the tissue to adhere. For the fresh frozen samples, 5 ml of fixation buffer (4% paraformaldehyde in 1x Phosphate buffered saline) were added in a fume hood and incubated for 15 min at room temperature. The fresh frozen samples were then washed with 5 ml Phosphate buffered saline 3 times, 5 min each.

Both the fresh frozen and the fixed frozen samples were then permeabilized in 5 ml 70% ethanol at 4 °C overnight, in parafilm-sealed dishes, and stored long-term in the same conditions. For hybridizing with the library (the gene panel), the samples were washed with 5 ml Vizgen Sample Prep Wash Buffer (Vizgen part number 20300001) and then incubated in 5 ml Formamide Wash Buffer (Vizgen pn 20300002) at 37 °C for 30 min in an incubator. The Formamide Wash Buffer was aspirated from the tissue, and 50 µl of the gene panel mix was added on top of each tissue. A piece of parafilm ~1.5 cm×1.5 cm was placed on top to spread the library mix and protect it from evaporation. The dishes were sealed with parafilm and placed in a humidified incubator at 37 °C for 36–48 h. The parafilm was removed from the top of each tissue, and the samples were incubated in 5 ml Formamide Wash Buffer at 47 °C for 30 min, twice. The samples were then washed with 5 ml Sample Prep Wash Buffer for 2 min.

**Gel embedding.** To gel embed the samples, fresh 10% w/v ammonium persulfate solution was prepared. For each sample, 5 ml of Gel Embedding Premix (Vizgen pn 20300004) was combined with 25 µl of the 10% ammonium persulfate solution and 2.5 µl of TEMED (N,N,N′,N′-tetramethylethylenediamine). In parallel, one 20 mm Gel Coverslip (Vizgen pn 20400003) for each sample was cleaned with RNAseZap, followed by 70% ethanol and dried with Kimwipes. The

Gel Coverslips were then covered with 100 µl of Gel Slick Solution (VWR, catalog number 12001-812) for a minute and wiped dry with Kimwipes. The Sample Prep Wash Buffer was aspirated from the samples. For each sample, 100 µl of the Gel Embedding Mix was retained in a small tube, while the remainder of the Gel Embedding Mix was added to the samples and incubated for 1 min. The Gel Embedding Mix was then poured out from the samples into a waste tube but kept aside on the bench (to monitor gel formation). The slides were then aspirated dry, leaving just enough liquid to keep the tissue from drying out. 85 µl of the separately retained Gel Embedding Mix was added on top of the tissue, and the Gel Slick treated coverslip was placed over it with tweezers, with the Gel Slick-treated side facing down toward the tissue and avoiding air bubbles. Extra Gel Embedding Solution was aspirated from the sides of the coverslips. The dishes were incubated at room temperature for 1.5 h to allow the gels to form. Thereupon, the coverslips were removed using a Hobby Blade and tweezers.

**Tissue clearing.** To clear the samples of lipids and proteins that interfere with imaging, 5 ml of Clearing Premix (Vizgen pn 20300003) were mixed with 50 µl of Proteinase K for each sample. After the coverslips were removed from the gel embedded samples, the clearing solution was added to each sample, and the dishes were sealed with parafilm. The fresh frozen samples were placed at 37 °C in a humidified incubator overnight, while the fixed frozen samples were placed at 47 °C in a humidified incubator overnight (or for a maximum of 24 h), and then moved to 37 °C. The samples were stored in the Clearing solution in the 37 °C incubator prior to imaging for up to a week.

**Sample imaging.** The Clearing solution was aspirated from the sample, and the sample was washed three times with Sample Prep Wash Buffer briefly, then again for 10 min on a rocker, and then three more times briefly. The sample was incubated with 3 ml of the appropriate first hybridization buffer, including Dapi and polyT reagent (Vizgen pn 20300021), for 15 min at room temperature on a rocker, covered from light. The sample was then washed with 5 ml of the Formamide Wash Buffer (Vizgen pn 20300002) for 10 min at room temperature on a rocker, covered from light, and then transferred to 5 ml of the Sample Prep Wash Buffer (Vizgen pn 20300001). In the meantime, the Imaging buffer was prepared by combining the Imaging buffer, the Imaging Buffer Activator (Vizgen pn 20300015), and RNase inhibitor at a ratio of 500:2.5:1. The hybridization buffers appropriate to the gene panel, as well as the imaging buffers, were loaded onto the Vizgen microscope system. The sample was placed in the flow chamber and connected to the fluidics system of the Vizgen microscope, taking care to disperse air bubbles. A low-resolution mosaic was acquired using a 10X

objective, and the regions of interest (whole coronal sections) were selected for high-resolution imaging with a 60x lens.

For the high-resolution imaging, the focus was locked to the fiducial fluorescent beads on the coverslip. Seven 1.5 μm-thick z planes were taken for each field of view when imaging the tissue, including for the DAPI channel. Cell segmentation was performed using the Watershed algorithm, using DAPI nuclear seeds and PolyT total RNA staining basins. Images were decoded to RNA spots with xyz and gene id using Vizgen's Merlin software.

### MERFISH analysis

**Data quality control and filtering.** Single-cell gene expression matrices were obtained by counting mRNA molecules within segmented cell boundaries and further analyzed in R using Seurat package and custom-made scripts. We excluded cells containing <30 or >2500 individual transcripts, <5 unique genes, cells with volume <40 $\mu m^3$ or >2500 $\mu m^3$ or cells with average count of blank probe spots >1.

**Annotation of cell types and clustering analysis.** Data were normalized by dividing gene counts for each cell by total count for that cell, multiplied by 10,000 and log-transformed. Data were then scaled and principal components were calculated on all 287 measured genes. Between 20 and 30 principal components were used to calculate UMAP embedding and perform clustering analysis using Louvain algorithm to examine each section individually. Afterwards, we integrated data from three replicates using Seurat's rpca workflow and repeated UMAP and clustering analysis. Cluster markers were identified using Wilcoxon rank-sum test. To resolve neuronal subclasses, we downloaded Allen Brain Atlas reference single-cell RNA-Seq data, including metadata with cell type annotations[25]. We then subsetted the reference data to neuronal classess and mapped cell type annotations to our integrated MERFISH data using Seurat's label transfer workflow. These annotations as well as annotation of remaining cell types (striatal neurons, glial, immune cells) were refined in rounds of subclustering analysis by each time subsetting data to major cell class and repeating normalization, scaling, pca and clustering workflows. During analysis we noted that fraction of segmented single-cell profiles exhibits contamination with transcripts from other cell types originating mostly from imperfect cell boundary segmentation. These resemble "doublets", however unlike doublets in droplet-based single-cell RNA-Seq workflows, in MERFISH data they bear biological significance as they originate from physically proximal cells. Therefore, we opted to annotate clusters that clearly represented mixture of different cell types as doublets, but keep them in the data, keeping this in mind during analyses. During subclustering analysis, we further removed effect of contaminating RNA by regressing expression signatures of other cell types from the data. For this, we first obtained signatures of each cell type by searching for differentially expressed genes between analyzed cluster and other coarse clusters (other present cell types) using strict threshold. We then calculated aggregate score for each signature in each cell using Seurat's AddModuleScore function and regressed these scores from expression matrix of analyzed cluster before running PCA, UMAP and subclustering analyses. For analysis of astrocytes, we iteratively repeated the subclustering in 3 rounds, each time removing the smaller subclusters that clearly exhibited profiles mixed with other cell types. For subclustering of microglia, we further excluded known non-microglial genes (known markers of other cells) from the matrix. This strategy dramatically improved resolution of clusters and allowed us to discover expression patterns previously masked by contaminating RNA while keeping sufficient cell numbers.

**Pseudotime analysis.** Pseudotime within microglia was calculated using Monocle3 (package version 1.3.1). The Seurat object with 3D UMAP embeddings and cluster information (res 1.5) was converted into a cell data set object using SeuratWrapper functions and used as input for trajectory inference. Principal graphs were calculated with learn_graph without considering partitions or closed loops, followed by calculating pesudotime with order_cells (selected root state within the homeostatic subpopulation). Pseudotime values for each single cell were then used to explore pseudotime per subcluster or within the spatial context.

**Spatial segmentation of the lesion.** To quantify cell types within the lesion areas, we in silico dissected lesion areas into lesion core, lesion inner edge and lesion outer edge. First, we segmented area representing lesion core based on: (i) spatial pattern of microglia (accumulated in lesion core, decreasing density towards lesion edge), (ii) spatial pattern of oligodendrocytes (absent in lesion, marking lesion edge) and (iii) expression profile of Mbp (marking lesion edge). We then expanded the polygon around lesion core twice by 50 um, segmenting inner lesion edge and outer lesion edge. For comparison, we also segmented area of uninjured white matter and uninjured cortical gray matter from contralateral hemisphere. We then quantified proportions of cell types in these areas.

**Cell neighborhood analysis.** To analyze cellular neighborhoods of microglial clusters and T-cells in the lesion, we first subsetted data to union of cells from lesion core, lesion inner edge and lesion outer edge. We then identified each cell's 5 nearest neighbors in physical space (based on Euclidean distance) and counted the proportion of each cell type label among the neighbors. We then repeated this process 10 000 times, each time randomly permuting cell type labels to obtain empirical *p*-value and empirical fold-enrichment (defined as observed fraction of cell type label among neighbors divided by average fraction obtained from all permuted iterations). Neighborhood analysis of the EM data was performed in the same way using EM-derived annotations and cellular positions.

**Spatial interaction analysis.** To evaluate spatial interactions between cell types in the lesion, we calculated Ripley's L function (specifically Besag's transformation of Ripley's K function) for pairs of cell clusters, using Lcross function from spatstat R package. L function is a cumulative statistic measuring average number of cells (points in 2d area) that are closer to a given cell than a certain distance. For plotting, we centered *L*-values by subtracting the measured value from the expected value (average of 200 random permutations of cell labels). Thus, values >0 indicate attraction/clustering between cells and values <0 indicate repulsion at a given distance. Values outside of the envelopes around L-function indicate statistical significance. We considered only the area of the lesion (bounded by "lesion outer edge" polygon) and evaluated L-value at distances from 5 um to 250 um.

**Alignment of MERFISH and EM data.** For initial exploration, DAPI staining images of MERFISH sections were registered onto EM overview scans of the sections of the same sample based on user-defined anatomical landmarks with BigWarp ImageJ plugin using thin plate spline transformation. After obtaining EM image stacks of the lesion, performing single-cell morphological analysis and recording positions of cells in the lesion core in the image coordinates, we repeated the alignment of the lesion area to the MERFISH data. Specifically, we identified corresponding landmark points in the EM stack image of the lesion area and MERFISH DAPI image, and then used icp function from Morpho R package to register the landmark points through iterative closest point algorithm. We performed this process in steps starting with rigid registration followed by similarity registration to align global coordinates and finally, thin-plate-splines registration (using tps2d function) to refine the alignment by correcting local deformations.

**Analysis of similarity between cellular spatial patterns.** EM and MERFISH data in common coordinate system were used for the analysis. For each cell cluster from both modalities, cellular density in the

area encompassing the registered cells was calculated using density function from spatstat R package. We calculated densities at multiple smoothing scales by varying parameter sigma (sigma = 25/50/75/100). Density distributions were then normalized to 1 and Earth Mover's Distance (EMD) was calculated between pairs of clusters. EMD evaluates dissimilarity of two probability density distributions by measuring the amount of "work" (represented by the amount of probability mass and the distance it needs to be moved) needed to transform one distribution into the other. This distance metric has several desirable properties for global comparison of spatial patterns, including considering the overall layout and relative positioning of the density distribution bins, rather than values at individual bins only. Finally, the average of EMDs from four sigma scales was calculated to obtain a multiscale EMD.

**Imputation of unmeasured genes in MERFISH data.** Anchors between MERFISH microglia data and SmartSeq2 scRNA-Seq data were identified using Seurat's FindTransferAnchors function based on the same gene set as previously for clustering (excluded non-microglial genes) and using rpca reduction with 15 dimensions. Subsequently expression of unmeasured genes was imputed using Seurat's TransferData function. We filtered out genes with low expression (requiring at least 10 counts in at least 20 cells based on original SmartSeq2 data) resulting in imputed expression of 8687 genes in 15830 MERFISH microglial cells.

**Correlation analysis between structural features and gene expression.** EM and MERFISH data in common coordinate system were used for the analysis. From MERFISH data, we considered only cells withing the area surrounding the available EM data. We then smoothened the data by applying spatial distance-weighted averaging in order to (i) increase signal-to-noise ratio, (ii) compensate for potential misalignment and (iii) spatially transfer the values between the two modalities. Specifically, for each cell, we calculated 2d kernel-weighted average of the feature values of its neighbors (both gene expression based on neighboring MERFISH cells and structural features based on neighboring EM cells). For gene expression modality, we used the measured values for the genes that were present in MERFISH panel, and the imputed values for the remaining genes. The weights for averaging were defined by bisquare kernel function with maximal distance parameter 75 microns, beyond which the weight is zero. This resulted in 2d-smoothed gene expression values as well as 2d-smoothed structural feature values in all cells in the analyzed area in the lesion. Finally, we calculated spearman correlation coefficient for all gene-structural feature pairs. We identified enriched Gene Ontology Cellular Component terms among the top 200 correlating genes using enrichR package.

**Tissue collection and preparation for scRNA-Seq**
The mice (18 and 24 weeks old) were deeply anesthetized by 10 mg/ml ketamine and 1 mg/ml xylazine solution i.p. and perfused with cold HBSS between 9am-11am (to decrease circadian fluctuations). Each brain was removed and under a dissection microscope individually micro-dissected; gray matter was isolated from the frontal cortex and white matter form optic tract, medial lemniscus and corpus callosum (attached gray matter and choroid plexus were carefully removed). We used a microglia isolation protocol we previously described[26], that prevents ex-vivo transcription and automatizes the mechanical isolation parts using GentleMacs with the Neural Tissue Dissociation Kit (Papain) (Miltenyi Biotec). We added actinomycin D (Act-D, Sigma, No. A1410) to a final concentration of 45 μM into the dissociation solution and enzyme mix to prevent ex-vivo transcription. The dissociated cell suspension was passed through a 70 μm cell strainer (Corning, 352350) before labeling. Subsequently, cells were blocked with mouse FcR-blocking reagent (CD16/CD32 Monoclonal Antibody, eBioscience cat:14-0161-82,1100) and then stained with the antibody against CD11b

(PE/Cy7,M1/70, eBioscience, Cat:48-0451-82,1:200) and washed with PBS (Sigma, D8537). Cells were then resuspended in PBS with BODIPY 493/503 (1:2,000 from a 1mgml−1 stock solution in DMSO) and actinomycin D (final concentration of 45 μM) and incubated for 10 min at 37 °C. Cells were washed two times with FACS buffer. Cells were excluded by staining with DAPI (1:10,000; Invitrogen). Viable (DAPI negative) single immune cells (CD11b positive cells) were sorted by flow cytometry (SH800; Sony), and index-sorting mode was used to save BODIPY fluorescence values for each cell. Single-cells were sorted into 96 well plates filled with 4 μL lysis buffer containing 0.05% Triton X-100 (Sigma), ERCC (External RNA Controls Consortium) RNA spike-in Mix (Ambion, Life Technologies) (1:24000000 dilution), 2.5 μM oligo-dT, 2.5 mM dNTP and 2 U/μL of recombinant RNase inhibitor (Clontech) then spun down and frozen at −80 °C.

**scRNA-Seq library preparation and sequencing**
The 96-well plates containing the sorted single cells were first thawed and then incubated for 3 min at 72 °C and thereafter immediately placed on ice. To perform reverse transcription (RT) we added into each well a master mix of 0.59 μL H2O, 0.5 μL SMARTScribe™ Reverse Transcriptase (Clontech), 2 μL 5x First Strand buffer, 0.25 μL Recombinant RNase Inhibitor (Clontech), 2 μL Betaine (5 M Sigma), 0.5 μL DTT (100 mM) 0.06 μL MgCl2 (1 M Sigma), 0.1 μL Templateswitching oligos (TSO) (100 μM AAGCAGTGGTATCAACGCAGAGTACrGrG+G). Next RT reactions were incubated at 42 °C for 90 min followed by 70 °C for 5 min and 10 cycles of 50 °C 2 min, 42 °C 2 min; ending with 70 °C for 5 min for enzyme inactivation. Preamplification of cDNA was performed by adding 12.5 μL KAPA HiFi Hotstart 2x (KAPA Biosystems), 2.138 μL H2O, 0.25 μL ISPCR primers (10 μM, 5′ AAGCAGTGG-TATCAACGCAGAGT-3′), 0.1125 μL Lambda Exonuclease under the following conditions: 37 °C for 30 min, 95 °C for 3 min, 23 cycles of (98 °C for 20 s, 67 °C for 15 s, 72 °C for 4 min), and a final extension at 72 °C for 5 min. Libraries were then cleaned using AMPure beads (Beckman-Coulter) cleanup at a 0.7:1 ratio of beads to PCR product. Libraries were assessed by Bio-analyzer (Agilent 2100), using the High Sensitivity DNA analysis kit, and quantified using Qubit's DNA HS assay kits and a Qubit 4.0 Fluorometer (Invitrogen, LifeTechnologies). Further selection of samples was performed via qPCR assay against ubiquitin transcripts Ubb77 (primer 1 5'-GGAGAGTCCATCGTGGT TATTT-3' primer 2 5'-ACCTCTAGGGTGATGGTCTT-3', probe 5'-/5Cy5/ TGCAGATCTTCGTGAAGACCTGAC/3IAbRQSp/−3') measured on a LightCycler 480 Instrument II (Roche). Samples were normalized to 160 pg/μL. Sequencing libraries were constructed by using in-house produced Tn5 transposase. Libraries were barcoded, pooled and purified in 3 rounds of AMPure bead (Beckman-Coulter) cleanup at a 0.8:1 ratio of beads to library. Libraries were then sequenced with 100 bp paired-end sequencing on DNBSeq platform (BGI group) to a median depth of $8.6 \times 10^5$ reads/sample.

**scRNA-Seq analysis**
Demultiplexed Fastq files were quality-controlled with FastQC and reads were then aligned using rnaSTAR to the GRCm38 (mm10) genome with addition of ERCC spike-in sequences. To obtain single-cell gene expression matrices reads were counted with rnaSTAR using parameter "quantMode GeneCounts" and unstranded argument. Further analysis was performed in R using Seurat package and custom-made scripts. Samples were filtered for quality with several QC thresholds (Fig. S4b). Data from LPC-injected and control mice were integrated together using Seurat's CCA integration workflow: Expression was normalized by dividing gene counts for each cell by total count for that cell, multiplied by 10,000 and log-transformed, 3000 variable features were identified with SelectIntegrationFeatures() function, transfer anchors were found using FindIntegrationAnchors() function and data were integrated with IntegrateData() function. Data were scaled before calculating PCA and UMAP embedding using 30

PCs. Cell type clusters were identified using Leiden algorithm and annotated based on canonical cell type markers identified by Wilcoxon rank-sum test. After first round of coarse clustering, microglia were isolated and analyzed separately by repeating aforementioned workflow. Microglial cluster labels were then mapped from MERFISH data onto SmartSeq2 data by finding anchors using Seurat's FindTransferAnchors() function and mapping labels with TransferData() functions. Cluster markers were identified using Wilcoxon rank-sum test.

**Functional enrichment analysis.** Gene expression signatures for each microglial population were identified by (i) finding sets of differentially upregulated genes between said population and every other population (wilcoxon rank-sum test) (ii) intersecting these sets. Identified signatures are available in Supplementary Data 3. Enriched KEGG pathways and Gene Ontology terms in these signatures were then identified using Enrichr package. Full results are available in Supplementary Data 4. For analysis of published microglial signatures, marker genes were collected from relevant publications. Specifically, for lipid-associated macrophage signature we intersected markers of lipid-associated macrophages in adipose tissue[42], atherosclerotic aorta[43] and liver[44]. For DAM signature, we intersected markers of disease-associated microglia (DAM)[15], activated-response microglia (ARM)[17], and neurodegeneration-related microglia signature[67] and excluded genes in lipid-associated macrophage signature to assure specificity. For homeostatic signature we intersected markers of homeostatic microglia from refs. 15,67. For interferon signature we intersected signature of interferon microglia[67] and core module of interferon-stimulated genes[68]. Activity of each signature was scored in each cell using AddModuleScore() function. Signatures from literature are available in Supplementary Data 5.

**Analysis of BODIPY staining data.** Continuous BODIPY staining values were matched to the single-cell transcriptomes from Smart-Seq2 dataset based on the index-sorting information. Exploratory analysis revealed moderate batch effect in BODIPY data between control and LPC-treated samples. To remove any potential batch influence, we used only cells from LPC-treated samples in the subsequent analysis. BODIPY values were log10-transformed and differences between microglial clusters were tested with ANOVA followed by pairwise-post hoc tests using emmeans R package. To robustly identify correlations between genes and BODIPY values, we employed metacell approach that collapses transcriptionally highly similar cells into small groups. This mitigates sparsity and noise inherent in single-cell data that are known to hamper correlation analyses[46]. Metacells were constructed with ConstructMetacells function from hdWGCNA R package[47,69] using $k = 20$. We then calculated Pearson correlation coefficients between average values of genes and average BODIPY value in the metacells. To identify pathways and biological processes enriched in BODIPY-correlating genes, we used Gene Set Enrichment Analysis (via fgsea R package) with pearson correlation values as rank metric (Supplementary Data 6). Gene Ontology, KEGG and Hallmark Signature gene sets were retrieved from the Molecular Signature Database using msigdbr R package.

## Serial section electron microscopy using automated tape-collecting ultramicrotomy (ATUM)

Mouse cryotome sections adjacent to the ones analyzed by spatial transcriptomics were sectioned at 10 μm thickness and collected onto glass slides (SuperFrost Plus Gold, Thermo). Tissue sections stayed adherent to the glass slides during the processing for EM and were kept in slide containers (Simport™ Scientific LockMailer™ Tamper Evident Slide Mailer, Fisher Scientific). These holders were positioned in a wrack on an orbital shaker during the incubation steps and reagents were exchanged by pouring or pipetting. We postfixed the sections in 2.5% glutaraldehyde (Science Services) in 0.1 M sodium cacodylate (Science Services) buffer (pH 7.4) for 30 min and applied a standard rOTO protocol starting with 1 h incubation in 1% osmium tetroxide (Science Services), 1% potassium ferricyanide (Sigma) in 0.1 M sodium cacodylate buffer (pH 7.4). After washing and reaction with 1% thiocarbohydrazide (Sigma) for 20 min at 40 °C we applied a second osmium step (1% osmiumtetroxide in water, 1 h). The tissue was further contrasted in 1% aqueous uranyl acetate at 4 °C over night. Samples were dehydrated in an ascending ethanol series and infiltrated with LX112 in acetone (LADD). Large gelatin capsules covering the entire brain coronal section (size 000, 9.55 mm diameter, Science Services) were positioned onto the glass slides and cured for 2d at 60 °C. In order to remove the encapsuled sample from the glass slide we notched the resin around the tissue section with a sharp blade. The glass slide was then submerged into liquid nitrogen for several seconds

## Table 2 | Comparison of protocol changes between standard ATUM and STcEM ATUM for EM

| | standard ATUM (Kislinger, et al. 2020) | STcEM ATUM | reasoning | impact |
|---|---|---|---|---|
| **fixation** | perfusion by 4% PFA, 2.5% glutaraldehyde in 0.1 M cacodylate buffer, pH 7.4 | perfusion by 4% PFA in PBS pH 7.4; post-fixation of cryosections in 2.5% glutaraldehyde in 0.1 M cacodylate buffer, pH 7.4 | glutaraldehyde would impact MERFISH analysis | lack of glutaraldehyde in perfusion fixative restricts ultrastructural preservation |
| **freezing** | no | 14 h 15% sucrose 5 h 30% sucrose Tissue-Tek® O.C.T.™ Compound embedding Freezing to −80 °C | freezing needed for cryosectioning | freezing sacrifices ultrastructural preservation |
| **tissue slicing** | vibratome sections | cryotome sections | freezing needed for MERFISH, thinner sections allow the inspection of similar regions by both methods | 10 μm thin sections are prone to puffing up and require special handling during further processing |
| **tissue dissection** | dissection, area usually <3×3 mm | no dissection, area ~8×10 mm (an entire coronal adult mouse brain section) | matching the information content of EM and MERFISH for later decision on the region of interest | challenging for tissue flatness during contrasting/embedding and section removal from glass slide |
| **contrasting container** | glass vial | microscope slide container | cryotome slice is adhered onto a glass slide | larger amount of reagents needed |
| **embedding mold** | <7 mm diameter gelatin capsule or silicone mold | 9.55 mm diameter gelatin capsule | large tissue area | ultramicrotomy holder has to be fitted by removal of inner metal blocks; lower stability during ultramicrotomy |
| **tissue retrieval after curing** | manual removal | freeze-thaw cycles | thin, large sections tend to break during retrieval from glass slide | fragility |
| **Ultramicrotomy** | 3–4 mm diamond knives | 4–8 mm diamond knives | blockface size | ultrathin or serial sectioning harder |

and heated up in a 60 °C water bath. These freeze-thaw cycles were repeated until the tissue block could be removed from the glass slide.

The block was trimmed at the empty resin end using a rotary tool (Dremel) in order to fit it into a standard sample holder. We trimmed the tissue end to generate an approximately 3 × 5 mm block face bearing the entire cortex and the ventricle using a trimming machine (TRIM2, Leica). After taking single large sections covering both hemispheres, the block was trimmed to the ipsilateral region only (2 × 3 mm) for the whole area annotation (Fig. 2). Sections (127) at 200 nm thickness were collected onto an ATUMtome (Powertome, RMC) using a histo knife (Diatome) and collected on freshly plasma-treated (custom-built, based on Pelco easiGlow, adopted from M. Terasaki, U. Connecticut, CT), carbon nanotube (CNT) tape (Science Services). CNT tape stripes were assembled onto adhesive carbon tape (Science Services) attached to a 4-inch silicon wafer (Siegert Wafer) and grounded by adhesive carbon tape strips (Science Services). EM micrographs were acquired on a Crossbeam Gemini 340 SEM (Zeiss) with a four-quadrant backscatter detector at 8 kV. In ATLAS5 Array Tomography (Fibics), we acquired the whole section at 200 nm and ipsi- and contralateral regions of interest at 20 nm lateral resolution. For the serial sections, we imaged the whole section overview at 200 × 200 × 200 nm resolution and selected the region of interest within a thickness of 13.2 μm (66 × 200 nm). Every second section (axial resolution 400 nm) was imaged at 20 × 20 nm lateral resolution. This resulted in three image stacks, one covering the full area of interest at 1.2 × 1.2 mm and two covering 0.5 × 0.5 mm. The large image stacks were exported as tiles, stitched and aligned using Fiji TrakEM2 and three VAST files generated from them.

For the whole area annotation (Fig. 2) we trimmed the block face to a 2 × 3 mm size covering the ipsilateral lesion site. Serial sections (127 at 200 nm thickness) were taken and collected onto tape. We imaged the whole section overview at 200x200x200 nm resolution and selected the region of interest within a volume of 13.2 μm (66 × 200 nm) thickness. Every second section (z resolution 400 nm) was imaged at 20 × 20 nm lateral resolution. This resulted in three image stacks, one covering the full area of interest at 1.2 × 1.2 mm and two covering 0.5× 0.5 mm. The large image stacks were exported as tiles, stitched using TrakEM2 and three VAST files generated from them.

## Expert cell type annotation

Annotation of cell types according to the ultrastructural morphology was performed in VAST[70] on all three datasets. The respective cell of interest was investigated along the entire stack thickness, screened for ultrastructural features (lipid droplet, lysosomal content, ER branching), categorized and flagged at the stack surface. The structural criteria for categorization are summarized in Table 3. The three VAST object files were exported and reassembled in Blender.

## Quantitative structural analysis

**Segmentation.** For quantitative feature-based classification of ultra-structural microglial morphologies we segmented the entire organellar inventory of examined cells. These segmentations are cell-type independent, only the extent of prevalence of lipid droplets is characteristic to (foamy) microglia. Specifically, microglia from 17 consecutive tissue sections were segmented using VAST in one of the three image stacks that covered the entire area. For each cell, the plane with the biggest cellular area within the stack was selected and its features were manually segmented. Exclusion criteria were: (i) cells with most of its volume outside the stack and (ii) cells with representative planes that were damaged from sample preparation. Nuclear eu- and hetero-chromatin were differentiated by separate segments. Further organelle labeling comprised lysosomes, mitochondria and ER. In addition, areas with lipid droplets or other complex phospholipid inclusions (hall-marks of different processing state of the phagocytosed lipidic material) were segmented and termed "lipid droplet-filled area". In addition, "organellar cytoplasm" and "empty cytoplasm" refer to cytoplasmic areas with or without any aforementioned organelles, respectively.

**Generating raw features data from segmentation.** Segmentations were exported with VAST tools as binary image tiles of 8192×8192 pixels. The following processing steps were carried out with the help of a user-defined script using Napari for visualization[71] Tiles were repatched and whole cell segmentations were indexed using connected component labeling. For each indexed cell, the exact location was obtained from its bounding box coordinates. Other ultrastructural segmentations were referred to the whole cell segmentations. For nucleus, heterochromatin, lysosomes, lipid droplets, mitochondria and other organelles, the ultrastructural area was expressed as total number of pixels. The length of endoplasmic reticulum ("ER length") was calculated as arc length using OpenCV[72]. Finally, the "empty cytoplasm area" was obtained by subtracting the number of pixels from all other segmentations from that of the whole cell segmentation. These were represented as a structured database, which was subsequently cleaned by removing stray pixels and any cells with unsegmented ultrastructure before undergoing further feature selection.

**Clustering analysis.** PCA was calculated based on all structural features (areas of segmented ultrastructures and their non-redundant ratios) and first 20 PCs were used to calculate shared nearest neighbor graph using Seurat's FindNeighbors function. The snn graph was then clustered by

## Table 3 | Overview of major ultrastructural criteria for expert classification

|  | criteria | references |
|---|---|---|
| **microglia in general** | darker cytoplasm, oval nuclei, characteristic heterochromatin pattern showing a peripheral heterochromatin ring and several larger patches throughout the nucleus, other morphological characteristics depend on the activation state | (Nahirney and Tremblay 2021)[34] (Savage, et al. 2018)[35] |
| **homeostatic** | small cytoplasmic compared to nuclear area | this paper |
| **reactive non-lysosomal** | larger cytoplasmic compared to nuclear area, extended ER partially with parallel sheets, several mitochondria, few lysosomes | this paper |
| **reactive lysosomal** | like reactive non-lysosomal, several lysosomes | (van Eijk and Aerts 2021)[39] (Du, et al. 2019)[38] |
| **foamy** | lipid accumulation, overload of the cytoplasm with lipid droplets and other states of phagocytosed lipidic material | (van Eijk and Aerts 2021)[39] (Berghoff, et al. 2021)[36] (Cantuti-Castelvetri, et al. 2018)[8] (Bosch-Queralt, et al. 2021)[12] |
| **T-cell** | characteristic heterochromatin pattern, low amount of organelles, relatively small cytoplasm (compared to nucleus) | (Hirsh, et al. 2007)[37] (Wacker, et al. 2015)[73] |
| **unknown** | nuclear morphology similar to T-cells, large cytoplasm without organelles, mitochondria only close to nuclear envelope region, not peripheral |  |
| **astrocyte** | contain dark granules (glycogen), nuclear membrane concave (star-like) (specific to this protocol), processes with filaments (GFAP filaments), bright cytoplasm | (Nahirney and Tremblay 2021)[34] (Cali, et al. 2019)[74] (Maxwell and Kruger 1965)[75] |

Louvain algorithm with FindClusters function. Data were visualized as UMAP, which was calculated based on 15 PCs using RunUMAP function.

**Method for retrieving corresponding manual label.** As expert classification and feature-based ultrastructural analysis were conducted independently, along with inadvertent folds in some sections that resulted in imprecise cross-section alignment, exact coordinates of a cell can vary slightly between both datasets. To match cells from expert classification and feature-based analysis to a common index, euclidean distances were calculated between every coordinate of the two datasets. For each expert classified cell, the shortest distance was selected where it was then assigned the corresponding cell index. Fisher exact test was used to compare significance of overlap between expert-assigned labels and cluster labels in 144 cells for which both labels were retrieved.

## Immunohistochemistry

To quantify PLIN2 and GPNMB/Galectin3 double positive microglia, brain sections (16 μm thick) from LPC injected mice (4 months old, 14 dpi) were permeabilized for 10 min in sodium citrate buffer (0.01 M, pH 6.0) at 80 °C, blocked in Blocking Buffer (2,5% BSA, 2,5% fish gelatin, 2,5% FCS in 1xPBS) and incubated with Plin2 (Novus Biological NB110-40877, 1:200, rabbit), Iba1 (Synaptic Systems 234 009,1:400, chicken), Gpnmb (biotechne, BAF2330, 1:200, goat) or Galectin3 (Novus Biological, NBP2-16590, 1:400, rat) in diluted blocking buffer (1:5 PBS) with 0,05% saponin (Sigma Cat.47036) overnight at 4 °C. After repeated washing, section were incubated with secondary antibodies (AlexaFluor™ 488, donkey anti-chicken Invitrogen # A78948, 1:1000, AlexaFluor™555 donkey anti-rabbit, Invitrogen #A-31572, 1:1000, AlexaFluor™ 647 donkey anti-goat, 1:1000 or AlexaFluor™ 647 donkey anti rat, # A48272, 1:1000) diluted in blocking buffer (1:5 in PBS) with 0,05% Saponin for 2 h at RT. Nuclei were stained with Hoechst (2 μg/ml Hoechst 33342). Optical sections were acquired with a confocal laser-scanning microscope (Zeiss LSM 900 AiryScan) using a 63x/1.46 Oil-objectives (Zoom factor 1.3–2.5). The z-step in z-stacks was kept at 0.8 μm.

To quantify T cells within and outside the lesion area, brain sections of LPC injected mice (14 dpi) were permeabilized in permeabilization buffer (0,5% Triton X-100 in PBS) for 30 min and blocked with AffiniPure FAB Fragment Donkey Anti-Mouse IgG (H + L) (715-007-003, Jackson ImmunoResearch, 1:100 in permeabilization buffer) for 1 h, followed by incubation with blocking buffer (5% goat serum in permeabilization buffer) for another hour. Primary antibodies against Iba1 (Synaptic Systems 234004, guinea pig, 1:500), Stat1 (Cell Signaling Technology, clone 14994 S, rabbit, 1:500) and Cd8a (biolegend, 100702, clone 53-6.7, rat, 1:100) were diluted in blocking buffer and incubated on the sections overnight at 4 °C. After washing three times with permeabilization buffer, sections were blocked using the Avidin/Biotin Blocking Kit (Vector's Laboratories, SP-2001) and then incubated with secondary antibodies against guinea pig (AlexaFluor™ 647, Invitrogen, A-21450, 1:500), rat (AlexaFluor™ 488, Invitrogen, A-11006, 1:500) and rabbit (Goat Anti-Rabbit IgG Antibody (H + L), Biotinylated BA-1000) in blocking buffer on the sections for 1 h. Sections were washed three times with permeabilization buffer and incubated with fluorescently conjugated streptavidin (Streptavidin, AlexaFluor™ 555 conjugate, Invitrogen, 1:500) in blocking buffer for 1 h. Lastly, DAPI (Invitrogen, D1306, 1 μg/mL final concentration) in blocking buffer was added for 30 min. Sections were finally washed and embedded in ProLong™ Gold Antifade Mountant (Life Technologies, P36934). Images were taken as z-stacks at a Zeiss LSM 900 AiryScan, with a 20x objective (Plan-Apochromat 20x/0,8 M27-Air).

**Analysis of IHC data.** For IHC quantification, images were taken from 3 to 5 biological replicates. The value of n and what it represents in each quantification can be found in the respective figure legends. For the quantification of double positive PLIN2+Galectin3+ cells, the average of 2–4 lesion side images per mouse was considered as one biological replicate. For the T cell count within and outside the lesion side, one whole imaged lesion and surrounding area per mouse was considered as a biological replicate. For the count of T cells outside of the lesion area, ventricles were excluded, as well as brain area without any further T cell occurrence, meaning the most distant T cell outside the lesion in the image frame built the landmarks for framing the outside lesion area, however, if not applicable (because of missing T cell), the border of the lesion side built the alternative landmark to frame the "outside lesion area". Quantification was performed with ImageJ. Bar graphs and boxplots related to IHC staining were generated in R and base R functions were used to perform statistical testing and to calculate p-values for Fisher's exact test or Wilcoxon-Mann-Whitney test. A p-value of <0.05 was considered as significant.

## Reporting summary

Further information on research design is available in the Nature Portfolio Reporting Summary linked to this article.

## Data availability

The MERFISH and scRNA-Seq datasets generated in this study have been deposited in the Gene Expression Omnibus (GEO) database under accession number GSE202638. Figure source data are provided with this paper. All other data that support the findings are available upon request from the authors. Source data are provided with this paper.

## Code availability

The code used for the analyses is available at Github: https://github.com/ISD-SystemsNeuroscience/STcEM.

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

## Acknowledgements

The work was supported by grants from Chan-Zuckerberg Initiative grant (to M.Si. and O.G.), Else Kröner-Fresenius-Stiftung grant (to O.G.) and the German Research Foundation (Project 458957955, TRR 274/1— ID 408885537- projects Z01, Z02, Munich Cluster of Systems Neurology, SyNergy, EXC 2145 SyNergy, ID 390857198) (to M.Si., M.Sc. and O.G.) and (ImmunoSensation—EXC2151–390873048) (to O.G.). For the single-cell and sorting studies, we are grateful for support from the "Flow Cytometric Cell Sorter Sony SH800 Core Unit" run by the Department of Vascular Biology at the Institute for Stroke and Dementia Research (ISD) and SyNergy EXC2145. the China Scholarship Council (CSC) fellowship program (to L.L. and H.J.). We thank Georg Kislinger, Cornelia Niemann and Birgit Kunke for technical assistance with EM. We also thank Natalia Petrenko and Jiang He for technical assistance with MERFISH.

## Author contributions

O.G. conceived and supervised the project. P.A., M.Sc., K.P.A., L.C.C., Han.J., Hao.J. L.L., G.G., S.A.B., S.B.-G., J.K., M.S., O.G., planned and performed experiments and analyzed the data. P.A., O.G. wrote the manuscript with input from all authors.

## Funding

## Competing interests

The authors declare no competing interests.
