## [Peer Review File · Nature Communications]

Spatial Transcriptomics-Correlated Electron Microscopy maps transcriptional and ultrastructural responses to brain injuryREVIEWER COMMENTS

Reviewer #1 (Remarks to the Author):

In this paper Androvic et al. present a new approach that integrates MERFISH transcriptomics with large area chip mapping using SEM. Correlating the molecular signature specificity of multiplex in situ hybridization with the ultrastructural detail of electron microscopy. This was achieved by selecting adjacent brain sections and processing them in parallel for transcriptomics and for electron microscopy, and being able to match them due to the reduced thickness of the sections. They applied this technology to a mouse model of brain injury, in which the authors were able to match the distinct transcriptomic signatures of different microglial states with different cells under electron microscopy. T cells were also examined in this model.

‘After demyelinating injury, microglia and macrophages proliferate and migrate into demyelinating lesions where they phagocytose and metabolize lipid-rich myelin, clear damaged myelin sheaths, initiate repair mechanisms and communicate with cells of the adaptive immune system⁷⁻⁹.’
Microglia proliferate and macrophages infiltrate. It is worth making that distinction to better reflect the biological events taking place in the injured brain.

As a follow up question, if the authors mention microglia and macrophages separately, is their technique able to discriminate between these two cell types? If yes, could this be further discussed, if not, could the reasons be explained. In Figure S2, microglia and macrophages appear separate but in the text the authors refer to them as myeloid cells, does this mean they have pooled them? By definition, microglia originate from the embryonic yolk sac and other macrophages which can be found in the brain from the bone marrow.

T-cells are mentioned in the text without context, are T-cells common in this type of model? It would be important to provide more information on T cells. The identification of T cells in this study is lacking experimental support as currently presented. The defining ultrastructural features of T cells are not discussed, while their markers which can also be expressed by microglia in some contexts would need to be clarified.

‘Blind to the MERFISH data, we annotated cells in the demyelinated area into categories according to their ultrastructural morphology in multiple EM sections (Figure 2D).’

The weakest part of the paper pertains to the lack of ultrastructural identification criteria provided and the low quality images not revealing the distinctive features of the identified cells. The authors mention they have used an ultrastructural classification for their groupings but they do not give any information on the classification guidelines. The authors also state to have done the classification independently of the transcriptomics data, but do not give the guidelines for the different cell groups in the text nor in the methods sections. It is difficult to judge from the few low-resolution pictures provided if this classification is valid without having information regarding the ultrastructural analysis process being disclosed. The whole correlation would fall through if the ultrastructural classification is not properly

done. Several examples from each cell types and states would need to be provided, considering how heterogeneous microglia are (even within a given state), with detailed identification criteria and zoom-in views at high-resolution on these distinctive features (e.g. pertaining to the nuclear heterochromatin, organelles, inter-cellular interactions, etc).

'EM annotation revealed multiple myeloid subgroups, including normal-appearing microglia, activated myeloid cells with high content of lysosomes, myeloid cells with activated morphology but low lysosomal content, foamy myeloid cells characterized by excessive deposition of lipid droplets in their cytoplasm and one other class of unknown immune cell type (Figure 2D).'

Was the myeloid nature of all the classifications confirmed solely on ultrastructure or was there another method of confirmation? The dark microglia, a microglial state identified at the ultrastructural level, could also be examined and discussed.

At the ultrastructural level, what do the authors mean by activated morphology? Of note, the field has moved away from this terminology considering that microglia are always active, in health and disease, varying their functions in a contextually-dependent manner. It would be best to use the reactive wording for this reason.

I would be curious to see some data on the consistency and % of match between the MERFISH signature and the ultrastructural categories. Since at the ultrastructural level there is a degree of diversity even within well characterized microglial state. It would be important to (i) determine the inclusion features and organellar signature for each category and (ii) the % of times that that matched the same proposed MERFISH signature.

If the authors identify in their text the unknown cells as IRM, I would add that tag to Figure 2. If someone sees the figure and then read unknown it feels incomplete, especially if the proposed classification is in the article.

In Figure 2D, I would suggest having the scale bar in each figure in a visible size with its corresponding measurement.

The density of homeostatic microglia in the electron microscopy sections appears very low (based on the purple dots that are supposed to represent them). It would be important to compare the findings in this study with other ultrastructural investigations of microglia.

Reviewer #2 (Remarks to the Author):

In this manuscript, Androvic, Schifferer et al. have integrated MERFISH spatial transcriptomics with whole tissue electron microscopy techniques to generate a next-generation spatial transcriptomics method. In brief, they have applied MERFISH and EM to adjacent 10um thin sections but independently

prepared using regular procedures for MERFISH and EM samples. Androvic, Schifferer et al. have also conducted scRNA-Seq on the second cohort of mice to validate the microglial clusters identified with MERFISH. Then they attempted to combine the MERFISH transcriptomic data with EM information and identified a morphologically unknown cell type in the EM images as IRM. I appreciate the ambitious purpose and impressive efforts of this study. This paper may potentially advance the field by providing a powerful technology. However, the authors should strengthen the current analysis and highlight what they have achieved. In fact, the manuscript is incomplete. I will list my concerns and suggestions as follows:

1. The current manuscript includes 2 major figures. The significance of the entire study mainly relies on the identification of cells undefined in EM images as IRM by MERFISH and scRNA-Seq. However, the authors could have performed nano-Gold techniques on those EM samples to verify their conclusions. Also, extended characterization of such a unique morphology of IRM in EM images and how it relates to the IRM transcriptomics could have been done. IHCs focusing on the proteins to verify the connections between MERFISH and EM could also be added.
2. I do not understand the necessity of applying MERFISH if the authors only need the rough spatial information of certain microglia clusters. RNA scope with several transcriptomic markers would provide similar information. Could the authors identify the trajectory from scRNA-Seq analysis in the MERFISH images? If so, that could be an important and novel identification of spatial transcriptomic trajectories.
3. The analyses on MERFISH and EM images are largely disconnected: the conclusions from MERFISH remain in the tissue scale while the analysis on EM data remains on basic morphological descriptions.
4. Many additional analyses could have been done to provide important information. For example, LAM/Foamy cells were spatially separated from the IRM-T cell pairs, why is that? Are there any mechanistic explanations that MERFISH plus EM can indicate, providing deeper understanding of pathogenesis?
5. I wonder if the authors could apply both MERFISH and EM imaging on the attached sides (e.g. MERFISH on anterior side and EM on posterior side) to truly link two imaging results together.
6. Minor point: The arrangement of figure panels could be optimized for better understanding. It is not straightforward to precisely find the small figures.

Reviewer #3 (Remarks to the Author):

The manuscript from Androvic and Schifferer et al. describes a correlation image analysis strategy to link the morphology from EM and transcriptome from MERFISH. This strategy allows authors to observe the ultrastructure of certain cell types, which I think shows a degree of expansion in the Spatial Transcriptomics field. The limitation is that ST and EM data are not from the same cells. Although, STcEM can only correlate cell types forming clusters in their niche, and cannot be used to characterize cell types sparsely distributed, for example, tumor infiltrated regions. The case study in the manuscript is well chosen, which increases the impact of the work. Addressing all the below points are necessary for publication.

Major points:

- 1) Morphological phenotypes from HE or other histological analyses are available in some ST methods

from the same cells, like 10x and NanoString. I would like to know why the authors choose EM rather than other morphological phenotypes. What are the potential application scenarios?

2) I saw MERFISH data from STcEM has fewer transcripts and genes detected. Why does this happen? Which modifications to the MERFISH protocol caused this problem? Does this cause problems for the cell types identification?

3) The selling point of the paper is to correlate the EM morphological information and transcriptional signatures. I feel showing differences in ultrastructure from different cell types could enhance this claim, like neurons, OPC, and oligodendrocytes. The EM figures in the paper should also be annotated well to assist the understanding of these EM images. Structures like lipid droplets, lysosomes, and nuclei should be labeled.

4) On page 4, "Spatial matching of remaining classes revealed strong concordance of patterns of homeostatic microglia, and activated microglia classes, demonstrating how intersection of morphological and molecular phenotype provided by STcEM can reveal tissue organization." Fig 2E didn't show Neighborhood analysis of Homeo and DAM.

5) Neighborhood analysis provides a statistical way to link morphological information and transcriptome. However, the reliability of this way is not validated by authors or others. Thus, I feel claims based on this analysis should be either adjusted or provide additional supporting data, like IHC data showing the co-localization of markers (lipid and Gpnmd).

6) Some description in the manuscript is not clear and should be adjusted. Cells defined as "unknown" with EM are too misty. Authors should provide a clear description of this cell type, like the lipid droplets number, lysosome number, or other intuitionistic features. Otherwise, they can be thought to be cells that cannot be defined by EM and represent multiple cell types. And I also think it's not enough to claim cells with Unknown structures are IRMs with Neighborhood analysis. More evidence or analysis details should be provided.

Others:

1) The scale bar annotation in Fig. 1 is missing.

2) The modified protocol steps should be indicated in the method section.

3) The detailed annotation criteria of cell types according to the ultrastructural morphology should be included in the method section, like how many lipid droplets are needed to define a foamy cell?

Response to reviewers:

General remarks:

We would like to express our gratitude to all three reviewers for their insightful and helpful comments that have greatly contributed to improving the quality and impact of our paper. We have considered all of the issues raised and have addressed them with new experiments and analyses to the best of our ability.

One of the key suggestions made by the reviewers was the need for more detailed information on the ultrastructural characterization of microglial classes. In response, we have substantially expanded this section of the manuscript by not only providing additional details and examples on the morphological classification, but also by adding a new quantitative analysis based on segmented subcellular structures. We were also able to identify correlations between the quantified ultrastructural features and gene expression. We are grateful for these suggestions, which have greatly improved the quality and relevance of our work.

Additionally, we have addressed the remaining comments and concerns raised by the reviewers. As a result, and as suggested by the editor, we expanded the study from a Brief Communication to a Full Article and due to this extensive change, we did not highlight text changes. Our manuscript thus now includes 6 main figures, 9 supplemental figures, and 3 tables. To facilitate the review process, we incorporate the manuscript text and figures into a single pdf and submit separate high-resolution figure files at sufficient resolution for reviewers to properly assess the data.

Overall, our revised manuscript introduces a novel method that is capable of resolving ultrastructural and transcriptional responses of single cells to brain injury, with potential applications across multiple fields. We sincerely hope that our efforts to address the reviewers' comments and concerns have resulted in a significant improvement to the manuscript. We greatly appreciate your consideration of our revised submission and look forward to your response.

REVIEWER COMMENTS

Reviewer #1 (Remarks to the Author):

In this paper Androvic et al. present a new approach that integrates MERFISH transcriptomics with large area chip mapping using SEM. Correlating the molecular signature specificity of multiplex in situ hybridization with the ultrastructural detail of electron microscopy. This was achieved by selecting adjacent brain sections and processing them in parallel for transcriptomics and for electron microscopy, and being able to match them due to the reduced thickness of the sections. They applied this technology to a mouse model of brain injury, in which the authors were able to match the distinct transcriptomic signatures of different microglial states with different cells under electron microscopy. T cells were also examined in this model.

After demyelinating injury, microglia and macrophages proliferate and migrate into demyelinating lesions where they phagocytose and metabolize lipid-rich myelin, clear damaged myelin sheaths, initiate repair mechanisms and communicate with cells of the adaptive immune system⁷⁻⁹.

Microglia proliferate and macrophages infiltrate. It is worth making that distinction to better reflect the biological events taking place in the injured brain.

Thank you for pointing this out. We have now changed the wording to the more neutral umbrella term “microglia accumulate at the injury site”.

As a follow up question, if the authors mention microglia and macrophages separately, is their technique able to discriminate between these two cell types? If yes, could this be further discussed, if not, could the reasons be explained. In Figure S2, microglia and macrophages appear separate but in the text the authors refer to them as myeloid cells, does this mean they have pooled them? By definition, microglia originate from the embryonic yolk sac and other macrophages which can be found in the brain from the bone marrow.

Thank you for your comment on our manuscript. We fully acknowledge that microglia and macrophages share many characteristics, making it challenging to differentiate them using bulk approaches. However, utilization of advanced techniques with single cell resolution, such as scRNAseq and MERFISH in our study allowed us to distinguish between these two cell types based on established marker genes with relatively high confidence (Jordao et al., 2019).

Our findings revealed a large population of microglia with higher expression of *Selp1g*, *Laptm5*, *Csf1r*, and *Tmem119*, and small population of CNS-associated macrophages (CAMs) identified through their expression of *Mrc1*, *Cd163*, and *Msr1*. These results are consistent with previous research that used genetic labeling to differentiate microglia from infiltrating macrophages in the context of LPC-induced demyelination in the CNS (Plemel et al., 2020).

The electron microscopy analyses, on the other hand, cannot readily discriminate between microglia and macrophages, which is why we relied on the information obtained from adjacent sections by MERFISH. This serves as another illustration of how our STcEM approach can assist in refining cell type identities in EM analyses. We have edited the nomenclature in the manuscript accordingly and discuss these important points in the Results section, with the data presented in the new Figure S3.

T-cells are mentioned in the text without context, are T-cells common in this type of model? It would be important to provide more information on T cells. The identification of T cells in this study is lacking experimental support as currently presented. The defining ultrastructural features of T cells are not discussed, while their markers which can also be expressed by microglia in some contexts would need to be clarified.

We appreciate the helpful suggestion to provide more information on T-cells. In the case of LPC-induced demyelination, T-cells show a brief influx in limited numbers, which has been shown to be influential, as the absence of T-cells in nude mice or the use of functionally blocking antibodies limits microglial activation and demyelination (Ghasemlou et al., 2007). In our recent study (Kaya et al., 2022), we found that during aging, a small population of T-cells can induce an interferon response in oligodendrocytes and microglia, leading to the loss of oligodendrocytes. We now refer these findings at appropriate places in the manuscript to provide context.

To better understand the role of T-cells, we selected genes in our MERFISH panel to discriminate T cells and other immune cells. The subclustering analysis of immune cell population in MERFISH data revealed small cluster of T-cells marked by *Ptpcr*, *Cd3e*, *Cd8a*, *Cd28*, *Cd247*, and *Ifng*, as shown in the new Figure S3. We further validated the presence of T-cells in LPC-lesions using immunocytochemistry of CD8a, as shown in new Figure S7D-E.

Regarding the ultrastructural features of T-cells, in our electron microscopy (EM) analyses, they can be identified by their nuclear heterochromatin pattern, small cytoplasmic versus nuclear area, and low amount of organelles (Hirsh et al., 2007, Wacker et al., 2015). We used these criteria in our manual expert EM analysis. Notably, a cluster of cells with similar characteristics is also captured by the newly added unsupervised clustering analysis based on

segmented structural features, as presented in Figure 4. These data are presented in Figures 2E-F, S3, and S7, and exemplify the role of T-cells in demyelination lesions.

‘Blind to the MERFISH data, we annotated cells in the demyelinated area into categories according to their ultrastructural morphology in multiple EM sections (Figure 2D).’ The weakest part of the paper pertains to the lack of ultrastructural identification criteria provided and the low quality images not revealing the distinctive features of the identified cells. The authors mention they have used an ultrastructural classification for their groupings but they do not give any information on the classification guidelines. The authors also state to have done the classification independently of the transcriptomics data, but do not give the guidelines for the different cell groups in the text nor in the methods sections. It is difficult to judge from the few low-resolution pictures provided if this classification is valid without having information regarding the ultrastructural analysis process being disclosed. The whole correlation would fall through if the ultrastructural classification is not properly done. Several examples from each cell types and states would need to be provided, considering how heterogeneous microglia are (even within a given state), with detailed identification criteria and zoom-in views at high-resolution on these distinctive features (e.g. pertaining to the nuclear heterochromatin, organelles, inter-cellular interactions, etc).

We agree with the reviewer that a proper ultrastructural classification is critical for the success of the method. To address this point, we provide more details on the ultrastructural criteria used by the expert in the main text and methods and we prepared a new table (Table 3), where we comprehensively summarize the classification criteria for each morphological category, along with literature references where appropriate. In addition, to complement and expand the expert analysis in an unbiased manner, we performed a new quantitative, feature-based analysis presented in new Figure 4. As described in the main text and methods, we segmented in 2D cellular structures including nuclear eu- and heterochromatin, lysosomes, mitochondria, endoplasmic reticulum, cytoplasm filled with lipid droplets, cytoplasm filled with organelles and “empty” cytoplasm. We quantified the area of each segmented compartment for 933 microglial cells in the lesion. We then performed clustering analysis based on these morphological features which revealed the presence of 5 clusters that largely overlap with the morphological categories from expert annotation as shown in Figure 4E. This also allowed us to quantitatively identify the structural hallmarks of each cluster, which we visualize in Figure 4C and 6C. As suggested by the reviewer, we provide more example images from each category with color-coded organellar structures in Figure 4B, along with their positions in morphological feature space that allows to appreciate the EM images in context of morphological heterogeneity of the whole dataset.

EM annotation revealed multiple myeloid subgroups, including normal-appearing microglia, activated myeloid cells with high content of lysosomes, myeloid cells with activated morphology but low lysosomal content, foamy myeloid cells characterized by excessive deposition of lipid droplets in their cytoplasm and one other class of unknown immune cell type (Figure 2D).’ Was the myeloid nature of all the classifications confirmed solely on ultrastructure or was there another method of confirmation? The dark microglia, a microglial state identified at the ultrastructural level, could also be examined and discussed.

The distinction of microglia from other cell types in EM is based on established knowledge as reviewed in (Nahirney and Tremblay, 2021). Their cytoplasm is darker compared to other cell types and the nuclei are oval with a characteristic heterochromatin pattern showing a peripheral heterochromatin ring and several larger patches throughout the nucleus (Savage et al., 2018). Depending on the activation state they contain parallel ER sheets, lysosomes and lipid droplets. The dark microglia, a microglial state identified at the ultrastructural level, is rarely present

under steady state conditions. It has been identified during chronic stress, aging, fractalkine signaling deficiency (CX3CR1 knockout mice), and Alzheimer's disease pathology (APP-PS1 mice) (Bisht et al., 2016) and we have observed this morphological type in our projects in Alzheimer's disease models (unpublished). However, in this demyelination model, we couldn't detect any typical characteristics of dark microglia including a darkened cytoplasm, stress markers (dilated ER, alterations in mitochondrial morphology) or changed heterochromatin pattern (Nahirney and Tremblay, 2021) (St-Pierre et al., 2020). We discuss this point in the revised discussion.

At the ultrastructural level, what do the authors mean by activated morphology? Of note, the field has moved away from this terminology considering that microglia are always active, in health and disease, varying their functions in a contextually-dependent manner. It would be best to use the reactive wording for this reason.

In our initial expert annotation we noticed increased ER and mitochondrial content and branching in some microglia that we originally called “activated”. In addition, some of these cells showed increased lysosomal content, which is ultrastructurally associated with high degradative burden (Du et al., 2019), and we originally classified these cells as “activated lysosomal”. We deduce that more ER and mitochondria would correspond to a more reactive state compared to a homeostatic microglial class but we agree this classification is not fully exhaustive or universally definitive. We changed the terminology to “reactive” microglia as suggested by the reviewer. While microglial classes have been described according to their transcriptional profile and overall shape (Heindl et al., 2018), sophisticated ultrastructural description is lacking, especially for the demyelination model presented here. Thus, characterization of the ultrastructural heterogeneity of microglia in this model is one of the contributions of our study.

I would be curious to see some data on the consistency and % of match between the MERFISH signature and the ultrastructural categories. Since at the ultrastructural level there is a degree of diversity even within well characterized microglial state. It would be important to (i) determine the inclusion features and organellar signature for each category and (ii) the % of times that that matched the same proposed MERFISH signature.

Thank you for this valuable feedback. As mentioned above, to address the point (i), we specified in detail the ultrastructural criteria considered for expert classification and summarize them in new Table 3. Additionally, we performed new quantitative analysis leading to new Figure 4. As described in the main text and methods, we segmented cellular structures including nuclear eu- and heterochromatin, lysosomes, mitochondria, endoplasmic reticulum, cytoplasm filled with lipid droplets, cytoplasm filled with organelles and empty cytoplasm and quantified the area of each segmented compartment for 933 microglial cells in the lesion. We then performed clustering analysis based on these morphological features, which revealed presence of 5 clusters that largely match the morphological categories from expert annotation as shown in Figure 4E. This also allows us to objectively identify the organellar signatures of each cluster, which we visualize in Figure 4C and 6C.

Regarding the point ii), it is unfortunately not possible to directly assess the match between the morphological and transcriptional signature for the same single cells, as the MERFISH and EM data are obtained from two different sections. Nevertheless, to quantitatively and objectively assess the matching between transcriptional (MERFISH-derived) and morphological categories, we utilized spatial coordinates as an anchor between the two datasets. We aligned MERFISH data and EM data to the common coordinate space as described in Methods, and then evaluated the correlation between spatial patterns of transcriptional clusters and both, expert-derived (Figure 3E-F) and clustering-derived morphological categories (Figure 4F) in the lesion area. As described in the main text and methods, we utilized two different metrics for

this evaluation – i) Earth Mover’s distance, which is a distance metric comparing the density distributions of cells from each category and ii) Ripley’s L function, which is a statistic measuring the overrepresentation of certain cell type in a radius of another cell type compared to random permutation labeling. These analyses provide a quantitative view on spatial matching between morphological and transcriptional categories and are in good agreement with our initial qualitative analysis. In particular, they confirm matching between foamy-ST cluster and foamy-EM class as well as IRM-ST and T-cell-ST clusters and IRM-EM and T-cell-EM morphological clusters.

If the authors identify in their text the unknown cells as IRM, I would add that tag to Figure 2. If someone sees the figure and then read unknown it feels incomplete, especially if the proposed classification is in the article.

Initially we named this morphological class “unknown” to stress the ability of the STcEM method to assign cell type/state label to the morphological class with distinct characteristics but unknown identity. We agree with the reviewer’s suggestion and changed the unknown cells to IRM. For the same reasons, we also changed the lipid-associated microglia (LAM) naming to foamy-ST.

In Figure 2D, I would suggest having the scale bar in each figure in a visible size with its corresponding measurement.

We changed accordingly.

The density of homeostatic microglia in the electron microscopy sections appears very low (based on the purple dots that are supposed to represent them). It would be important to compare the findings in this study with other ultrastructural investigations of microglia.

The electron microscopy characterization of microglia is limited to the lesion area where most microglia are reacting to the tissue damage, which explains low density of homeostatic microglia. The density of homeostatic microglia is also similarly low in adjacent MERFISH sections (Figure 4A-B). We appreciate the suggestion to compare our EM results to other ultrastructural investigations, however we are not aware of other studies that characterize microglial ultrastructure around LPC lesions in this detail. As mentioned in our previous response, we compared our results to the dark microglia phenotype found in models of AD, and we discuss this point in the revised discussion.

Reviewer #2 (Remarks to the Author):

In this manuscript, Androvic, Schifferer et al. have integrated MERFISH spatial transcriptomics with whole tissue electron microscopy techniques to generate a next-generation spatial transcriptomics method. In brief, they have applied MERFISH and EM to adjacent 10um thin sections but independently prepared using regular procedures for MERFISH and EM samples. Androvic, Schifferer et al. have also conducted scRNA-Seq on the second cohort of mice to validate the microglial clusters identified with MERFISH. Then they attempted to combine the MERFISH transcriptomic data with EM information and identified a morphologically unknown cell type in the EM images as IRM. I appreciate the ambitious purpose and impressive efforts of this study. This paper may potentially advance the field by providing a powerful technology. However, the authors should strengthen the current analysis and highlight what they have achieved. In fact, the manuscript is incomplete. I will list my concerns and suggestions as follows:

We thank the reviewer for the positive comments and helpful suggestions. We now substantially expanded our manuscript and strengthened the analyses to make it much more complete.

1. The current manuscript includes 2 major figures. The significance of the entire study mainly relies on the identification of cells undefined in EM images as IRM by MERFISH and scRNA-Seq. However, the authors could have performed nano-Gold techniques on those EM samples to verify their conclusions. Also, extended characterization of such a unique morphology of IRM in EM images and how it relates to the IRM transcriptomics could have been done. IHCs focusing on the proteins to verify the connections between MERFISH and EM could also be added.

The manuscript was originally written as brief communication which was limited to 2 figures. As suggested by the editor, we now extended the study from Brief Communication to a Full Article. As mentioned in our responses to Reviewer 1, to extend morphological characterization of not only IRMs, but all identified categories, we generated segmentation data and performed new quantitative analysis presented in Figure 4.

We agree with reviewer that bridging EM images to MERFISH and scRNA-Seq is a key part of our manuscript. In the revised version, we substantially strengthen this connection by registering EM and MERFISH data to common coordinate space and quantitatively evaluating the match between morphological and transcriptional categories. In addition, we challenged our method a step further and devised a strategy to correlate values of structural features to expression of individual genes, leading to a new Figure 6. This, among other findings, reiterated correlation between distinct structural features (e.g., ratio of mitochondria to other organelles, higher amount of empty cytoplasm) and expression of interferon-stimulated genes. We agree with the reviewer that identification of interferon-responsive microglia morphology is one of the significant contributions of our study. However, we would also like to highlight other findings, such as identification of transcriptional signature of microglia with foamy morphology.

As suggested by the reviewer, we extensively validate this connection by new data and analyses. The foamy-ST (originally called LAM) – foamy-EM link was based on their transcriptional identity, location and similarity of their neighborhood interactions in EM and MERFISH data. To provide more direct evidence of lipid droplet association with foamy-ST cluster, we combined single cell RNA-Seq information of CD11b+ cells with their lipid load by staining them with BODIPY, which is a lipid-binding dye that is commonly used to stain cellular lipid droplets. We found that foamy cluster has highest values of BODIPY signal (Figure 5E) and genes correlating with BODIPY values are largely foamy microglia signature genes (Figure 5F-G), validating our initial link between MERFISH and EM data. As an additional validation of this link, we performed IHC staining for lipid droplet-associated protein PLIN2 and top foamy microglia signature markers GPNMB and Galectin-3, which also confirmed significant association between these genes and lipid droplet-loaded microglia (Figure 5I-K). To validate IRM location next to T cells, we used STAT1 antibody which we used before to identify IRM microglia in white matter aging context (Kaya et al. 2022). Most cells in the LPC lesion were STAT1-positive (Figure S7) while the MERFISH identified sparse IRM presence. This difference may be explained by multi-gene interferon signature measured by MERFISH, while IHC has been limited to only one protein – STAT1. Alternatively, MERFISH measures short lived transcriptional interferon responses while Stat1 IHC measures the protein level responses that are more stable and persistent.

We believe this data and analyses represent an appropriate validation of our findings and substitute immunogold labeling that requires one concrete target antigen that would define a certain cell class. In contrast, IRM, foamy as well as other cell states are defined by a palette of

transcripts. On the technical side, immunogold labeling is only possible for very few antibodies and requires membrane permeabilization which reduces the ultrastructural quality. The most efficient immunolabeling technique for EM so far is the Tokuyasu method which results in an unconventional inverted ultrastructural image (Möbius and Posthuma, 2019; Petralia and Wang, 2021). The EM method we present here is also limited in ultrastructural preservation due to the requirements of the MERFISH protocol. The comparison of Tokuyasu and STcEM would result in completely diverging cell morphologies and hence, would not allow molecular annotation of STcEM ultrastructural data.

2. I do not understand the necessity of applying MERFISH if the authors only need the rough spatial information of certain microglia clusters. RNA scope with several transcriptomic markers would provide similar information.

RNA scope can be used to investigate only limited number of transcripts. In contrast MERFISH is a highly multiplexed imaging method that offers significantly higher throughput both in terms of number of investigated transcripts in parallel (reaching several hundreds) as well as tissue area that can be scanned in a short amount of time, thus representing a true single cell spatial omics technology. This allows to comprehensively identify not only coarse cell types, but also finer cellular states as well as their differentially expressed genes all in one analysis. Additionally, it allows MERFISH to be used not only as a validation method of preexisting signatures, but also as a discovery method. We demonstrate this by our analysis of microglia states responding to injury that are defined by multidimensional signatures. In the revised version of the manuscript, we further demonstrate this power by going beyond linking cell types. We correlated transcriptome-wide gene expression to structural characteristics of microglia (Figure 6). This was partly enabled by imputing missing gene expression from our single cell data, which is another advantage that MERFISH offers in comparison to RNA scope, which cannot facilitate this quantitative bridge to other omics modalities. However, we agree with the reviewer that STcEM approach is not necessarily bound to MERFISH and this technology could be replaced by similar alternatives, potentially probing additional omics modalities, which we mention in the revised discussion.

Could the authors identify the trajectory from scRNA-Seq analysis in the MERFISH images? If so, that could be an important and novel identification of spatial transcriptomic trajectories.

Thank you for the suggestion. We fully agree that understanding the relationships between the temporal dynamics of cellular processes and their spatial distribution is an exciting novel area of spatial biology. We have followed up on this comment, and used Monocle3 to calculate pseudotime ordering of microglia utilizing our MERFISH data (Figure S4). This analysis placed foamy microglia cluster at the end of pseudotime ordering, which is consistent with the fact that this cluster has the most significant downregulation of typical homeostatic microglia signature genes. When we plotted pseudotime values in space, we observed that cells in the lesion core had on average highest pseudotime values, indicating the degree of microglial reactivity is highest in this area. This is likely due to the relatively higher concentration of foamy cells in the lesion core. As the identification of spatial trajectories is rapidly developing field, more advanced computational approaches that also take spatial context into account may reveal additional insights in the future.

3. The analyses on MERFISH and EM images are largely disconnected: the conclusions from MERFISH remain in the tissue scale while the analysis on EM data remains on basic morphological descriptions.

As mentioned in the responses above, we significantly strengthened the connection between EM and MERFISH by new quantitative analyses. Specifically, we aligned EM and MERFISH data in a common space and compared the match between morphological and transcriptional categories (Figure 3D-F). We expand the expert analysis of ultrastructure by quantitative

analysis of segmented ultrastructural features (Figure 4). We also directly relate the two modalities by performing correlation analysis between structural features and gene expression (Figure 6).

4. Many additional analyses could have been done to provide important information. For example, LAM/Foamy cells were spatially separated from the IRM-T cell pairs, why is that? Are there any mechanistic explanations that MERFISH plus EM can indicate, providing deeper understanding of pathogenesis?

We hope that this reviewer will find that the newly added data and analyses provide examples of how the combination of MERFISH and EM can deepen our understanding of biology. To provide more information, we performed several new analyses. Apart from aforementioned linking of EM and MERFISH data, new analyses also include pseudotime analysis of microglia (Figure S4), or subclustering and spatial analyses oligodendrocytes (Figure S5) and astrocytes (Figure S6). With regards to the IRM-T cell pairs, our findings now demonstrate that the interferon response cell states are also induced in astrocytes and oligodendrocytes, that together with interferon-responsive microglia co-cluster with each other and occur in proximity of T cells.

5. I wonder if the authors could apply both MERFISH and EM imaging on the attached sides (e.g. MERFISH on anterior side and EM on posterior side) to truly link two imaging results together.

If we understood the reviewer correctly, this approach would not provide additional benefit because both MERFISH and EM data already probe almost the entire depth of the 10um thick cryosections. In MERFISH the thickness is probed by obtaining 7 z-stacks, each 1.5 um apart and the information from z-stacks is then merged during analysis. In EM data the thickness is probed by semithin sectioning (in our manuscript 200 nm apart) and morphology is investigated throughout these image stacks, with a technical note that in the processing steps for EM analysis the most superficial layers are lost, which would also prohibit true matching of the opposing sides of the MERFISH and EM data. That being said, we agree with the reviewer that obtaining the two imaging modalities from the same sections would be valuable advancement, and this represents a technical challenge we are working to optimize for future studies.

6. Minor point: The arrangement of figure panels could be optimized for better understanding. It is not straightforward to precisely find the small figures.

We have substantially expanded the manuscript from brief communication to full Article and in the process, we also re-arranged the figures and the text in hopes to better convey the information. We acknowledge that many of our figures are relatively data-heavy and may be complex, thus we have inserted on multiple occasions schematic figure panels visually summarizing experimental and analytical approach. We hope that this will aid in the understanding and make the figures more accessible to the readers.

Reviewer #3 (Remarks to the Author):

The manuscript from Androvic and Schifferer et al. describes a correlation image analysis strategy to link the morphology from EM and transcriptome from MERFISH. This strategy allows authors to observe the ultrastructure of certain cell types, which I think shows a degree of expansion in the Spatial Transcriptomics field. The limitation is that ST and EM data are not from the same cells. Although, STcEM can only correlate cell types forming clusters in their niche, and cannot be used to characterize cell types sparsely distributed, for example, tumor infiltrated regions. The case study in the manuscript is well chosen, which increases the impact of the work. Addressing all the below points are necessary for publication.

Major points:

1) Morphological phenotypes from HE or other histological analyses are available in some ST

methods from the same cells, like 10x and NanoString. I would like to know why the authors choose EM rather than other morphological phenotypes. What are the potential application scenarios?

Indeed, this is possible, and histological stainings paired with lower-resolution ST methods have been utilized in prior study to investigate the relationship between gene expression and morphology (He et al., 2020). However, these approaches are limited by the resolution of both, imaging and ST modalities, to investigations of crude cellular morphologies and bulk gene expression in tissue patches and tissue regions. We discuss this point and cite the study in the revised discussion.

Compared to histological analyses, EM is unprecedented in its resolution, allowing highly detailed imaging of cells, enabling simultaneous visualization of structure of the cells and organellar morphologies or contact sites. Furthermore, compared to fluorescence labeling techniques, it is untargeted, and the entire tissue morphology is retained which enables unbiased analysis of the entire biological context. Recent studies show that particularly subcellular organization of organelles is informative of fine-grained functional states (Cho et al., 2022; Gut et al., 2018; Viana et al., 2023). Histological staining cannot achieve this level of detailed information.

As pointed by the reviewer, histological analyses are currently available from the same tissue sections as lower-resolution, “spot-based” ST measurements with methods such as Visium (10x genomics), SlideSeq or GeoMX (NanoString). However, these ST methods are also dramatically lower in their resolution compared to MERFISH, which offers single cell, or even subcellular resolution. The difference in resolution afforded by EM+MERFISH combination is crucial for certain types of research, such as in neurodegeneration studies where changes in axonal and myelin morphology can indicate demyelination or neurodegeneration. We showcase the effectiveness of this combination by investigating relatively subtle cellular changes in both transcriptome and morphology that occur during pathology and their micrometer-scale spatial organizations (e.g. co-clustering of IRMs, IRAs, IROs and T-cells), findings that would be very difficult to discern with histological staining followed by current generation spot-based ST methods.

In addition, we provide new evidence of the utility of EM+MERFISH combination by demonstrating the ability to correlate individual subcellular features of cells to expression of individual genes, which we present in Figure 6. We see potential examples of application of STcEM beyond neuroscience in the fields of viral diagnostics and virology studies, where morphological criteria can be used to identify most known virus particles. Thus, STcEM could be utilized to identify viruses and infected cellular niches via EM, while simultaneously measuring cellular responses via MERFISH. Similar opportunities lie in renal biology, where EM ultrastructural measurements of the renal glomerulus, vasculature, and tubulointerstitial compartment remain a crucial part of the diagnosis of a number of disorders. We discuss these relevant points raised by reviewer in the discussion of the revised version.

2) I saw MERFISH data from STcEM has fewer transcripts and genes detected. Why does this happen? Which modifications to the MERFISH protocol caused this problem? Does this cause problems for the cell types identification?

The reviewer is correct. In the STcEM modified protocol, the tissues were PFA-fixed and then incubated for more than 26 hours in sucrose gradients prepared with non-RNase free solution due to requirements for EM imaging. It is therefore possible that some of the RNA could be degraded and some could be inaccessible compared to sections prepared from brains that were snap-frozen without additional manipulation. Despite these unfavorable conditions, and some degree of reduction in RNA detection rate, we obtained similar biological results as from the snap-frozen tissue section. We believe PFA crosslinking protected RNA from RNase activity

and preserved transcriptional profile of cells. As far as we could tell from our analyses, this partial loss of sensitivity did not impede the cell type identification or other analyses, as we found balanced representation of cell types and well-mixed populations in UMAP plot in all three sections as shown in Figures S2B and S2C. This is further supported by excellent linear correlation of relative gene abundances between all sections (Figure S1C). Notably, MERFISH as a single molecule technology displays very high sensitivity, estimated to reach over 90% (Moffitt et al., 2018), and therefore even some degree of reduction in RNA detection rate would not be expected to dramatically hamper analysis and biological insights.

3) The selling point of the paper is to correlate the EM morphological information and transcriptional signatures. I feel showing differences in ultrastructure from different cell types could enhance this claim, like neurons, OPC, and oligodendrocytes.

Our study focuses on LPC demyelinating lesions, which are primarily composed of microglia and T-cells, with a smaller presence of astrocytes. Other cell types are exceedingly rare, and we were only able to identify them using MERFISH that measured whole section, but not EM that imaged the lesion area.

We agree with the reviewer that bridging morphological information to the transcriptome would be valuable for all cell types and biological scenarios, and we have discussed this in the discussion section. As suggested by the reviewer, we have performed new quantitative analyses on the EM imaging data to strengthen our claims about the correlation of EM morphological information and transcriptional signatures. We have linked the two by matching morphological and transcriptional microglia clusters (Figure 4) and by identifying correlations between structural features and gene expression (Figure 6). Additionally, we have included validation of these significant links by staining for morphological and transcriptional markers (Figure 5I-K).

The EM figures in the paper should also be annotated well to assist the understanding of these EM images. Structures like lipid droplets, lysosomes, and nuclei should be labeled.

We agree with the reviewer's suggestion and we now show images with segmented cells (Figure 3D) as well as with segmented and color-coded organellar structures (Figure 4B).

4) On page 4, "Spatial matching of remaining classes revealed strong concordance of patterns of homeostatic microglia, and activated microglia classes, demonstrating how intersection of morphological and molecular phenotype provided by STcEM can reveal tissue organization." Fig 2E didn't show Neighborhood analysis of Homeo and DAM.

We apologize if the description of this figure wasn't clear. Figure 2E (in the revised version Figure 3C) did include the neighborhood analysis of homeostatic and DAM clusters, but the points are thresholded for significance and fold-change, therefore only interactions passing indicated p value and fold-change are visible as "bubbles", as shown in the panel legend. In the revised version, we complement this nearest-neighbor-based analysis with additional metrics correlating the spatial patterns of EM and MERFISH classes in common coordinates space, as also mentioned in our responses to reviewer #1. Specifically, i) Earth Mover's distance, which is a distance metric comparing the density distributions of cells from each category and ii) Ripley's L function, which is a statistic measuring the overrepresentation of certain cell type in a radius of another cell type compared to random permutation labeling. These analyses provide a quantitative view on spatial matching between morphological and transcriptional categories and are in good agreement with our initial analysis.

5) Neighborhood analysis provides a statistical way to link morphological information and transcriptome. However, the reliability of this way is not validated by authors or others. Thus, I feel claims based on this analysis should be either adjusted or provide additional supporting data, like IHC data showing the co-localization of markers (lipid and Gpnmd).

We agree with the reviewer that linking morphological information with transcriptome data is important. In the revised version, we strengthened this connection by aligning EM and MERFISH data in a common coordinate space and quantitatively evaluating the match between morphological and transcriptional categories as described in the answer above. As suggested by the reviewer, we also validated our findings on lipid-associated/foamy microglia by combining single cell transcriptome data of CD11b+ cells with their lipid load using the lipid-binding dye BODIPY. We found that the foamy cluster had the highest values of BODIPY signal, and genes correlating with BODIPY values were largely foamy signature genes, validating our initial link between MERFISH and EM data. To further support our conclusion, we have also included staining for lipid droplet-associated protein PLIN2 and top foamy signature genes GPNMB and Galectin-3 (encoded by Lgals3), as shown in Figure 5I-K. This revealed co-occurrence of these markers in Iba1+ microglia and quantification showed significant, almost 5-fold enrichment of Galectin-3-positivity in PLIN2-positive microglia compared to PLIN2-negative microglia.

6) Some description in the manuscript is not clear and should be adjusted. Cells defined as “unknown” with EM are too misty. Authors should provide a clear description of this cell type, like the lipid droplets number, lysosome number, or other intuitionistic features. Otherwise, they can be thought to be cells that cannot be defined by EM and represent multiple cell types. And I also think it’s not enough to claim cells with Unknown structures are IRMs with Neighborhood analysis. More evidence or analysis details should be provided.

We thank the reviewer for the feedback on the description of the "unknown"/IRM cell type. In the revised version, we have provided clear structural characteristics of the IRM (originally called “unknown”) class as well as all other classes in Table 3 and expanded and clarified the related sections of the main text and methods. To further support our findings, we performed new quantitative analysis presented in Figure 4. As described in the main text and methods, we segmented 8 cellular structures and quantified their area (together with their encompassing compartments) in EM image section for 933 microglial cells in the lesion. We then performed unsupervised clustering analysis based on these morphological features, which revealed presence of 5 clusters that largely overlap with the morphological categories from expert annotation as shown in Figure 4E, including a Cluster3 overlapping the IRM/unknown cell class, showing that these cells have relatively coherent and distinct morphology. We also performed spatial similarity analysis between EM and MERFISH classes that revealed that Cluster 3 matched the positions of IRM cluster. Furthermore, we unbiasedly identified the structural hallmarks of each cluster, including the Cluster 3, which we visualize in Figure 4C and 6C. This showed that Cluster 3 is marked by higher ratio of mitochondria to other organelles or higher ratio of empty cytoplasm to organelles-filled cytoplasm. We also provide more EM image examples from Cluster 3 as well as other clusters in Figure 4B. The visual examination of images from Cluster 3 reveals similar morphological characteristics as described by human expert (large cytoplasm without organelles, mitochondria only close to nuclear envelope region). Furthermore, we performed a new analysis, presented in Figure 6, that correlates values of structural features to expression of individual genes. This, among other findings, reiterated association between distinct structural features high in Cluster 3 and expression of interferon-stimulated genes marking the IRM cell state. Altogether, in the revised version we present clear structural characteristics and multiple lines of evidence for the link between the IRM morphology and IRM transcriptional state.

Others:

1) The scale bar annotation in Fig. 1 is missing.

Thank you for spotting this. We added scale bars accordingly.

2) The modified protocol steps should be indicated in the method section.

Thank you for the suggestion. We updated methods sections and also introduced new Tables 1 and 2 that compare standard protocol with STcEM-modified protocol from the perspective of both EM processing and MERFISH, and highlight the differences and their potential impact on handling and data quality.

3) The detailed annotation criteria of cell types according to the ultrastructural morphology should be included in the method section, like how many lipid droplets are needed to define a foamy cell?

We have prepared a new Table3, where we comprehensively summarize the classification criteria for each morphological category along with literature references when appropriate. As mentioned above, we also complement the expert analysis in an unbiased manner with a new quantitative analysis of segmented morphological features presented in Figure 4.

References:

- Bisht, K., Sharma, K.P., Lecours, C., Gabriela Sánchez, M., El Hajj, H., Miliot, G., Olmos-Alonso, A., Gómez-Nicola, D., Luheshi, G., Vallières, L., *et al.* (2016). Dark microglia: A new phenotype predominantly associated with pathological states. *Glia* 64, 826-839.
- Cho, N.H., Cheveralls, K.C., Brunner, A.D., Kim, K., Michaelis, A.C., Raghavan, P., Kobayashi, H., Savy, L., Li, J.Y., Canaj, H., *et al.* (2022). OpenCell: Endogenous tagging for the cartography of human cellular organization. *Science* 375, eabi6983.
- Du, H.-X., Chen, X.-G., Zhang, L., Liu, Y., Zhan, C.-S., Chen, J., Zhang, Y., Yu, Z.-Q., Zhang, J., Yang, H.-Y., *et al.* (2019). Microglial activation and neurobiological alterations in experimental autoimmune prostatitis-induced depressive-like behavior in mice. *Neuropsychiatric Disease and Treatment Volume 15*, 2231-2245.
- Ghasemlou, N., Jeong, S.Y., Lacroix, S., and David, S. (2007). T cells contribute to lysophosphatidylcholine-induced macrophage activation and demyelination in the CNS. *Glia* 55, 294-302.
- Gut, G., Herrmann, M.D., and Pelkmans, L. (2018). Multiplexed protein maps link subcellular organization to cellular states. *Science* 361.
- He, B., Bergenstrahle, L., Stenbeck, L., Abid, A., Andersson, A., Borg, A., Maaskola, J., Lundeberg, J., and Zou, J. (2020). Integrating spatial gene expression and breast tumour morphology via deep learning. *Nat Biomed Eng* 4, 827-834.
- Heindl, S., Gesierich, B., Benakis, C., Llovera, G., Duering, M., and Liesz, A. (2018). Automated Morphological Analysis of Microglia After Stroke. *Front Cell Neurosci* 12, 106.
- Jordao, M.J.C., Sankowski, R., Brendecke, S.M., Sagar, Locatelli, G., Tai, Y.H., Tay, T.L., Schramm, E., Armbruster, S., Hagemeyer, N., *et al.* (2019). Single-cell profiling identifies myeloid cell subsets with distinct fates during neuroinflammation. *Science* 363.
- Kaya, T., Mattugini, N., Liu, L., Ji, H., Cantuti-Castelvetri, L., Wu, J., Schifferer, M., Groh, J., Martini, R., Besson-Girard, S., *et al.* (2022). CD8(+) T cells induce interferon-responsive oligodendrocytes and microglia in white matter aging. *Nat Neurosci* 25, 1446-1457.
- Möbius, W., and Posthuma, G. (2019). Sugar and ice: Immunoelectron microscopy using cryosections according to the Tokuyasu method. *Tissue and Cell* 57, 90-102.
- Moffitt, J.R., Bambah-Mukku, D., Eichhorn, S.W., Vaughn, E., Shekhar, K., Perez, J.D., Rubinstein, N.D., Hao, J., Regev, A., Dulac, C., *et al.* (2018). Molecular, spatial, and functional single-cell profiling of the hypothalamic preoptic region. *Science* 362.

- Nahirney, P.C., and Tremblay, M.-E. (2021). Brain Ultrastructure: Putting the Pieces Together. *Frontiers in Cell and Developmental Biology* 9.
- Petralia, R.S., and Wang, Y.-X. (2021). Review of Post-embedding Immunogold Methods for the Study of Neuronal Structures. *Frontiers in Neuroanatomy* 15.
- Plemel, J.R., Stratton, J.A., Michaels, N.J., Rawji, K.S., Zhang, E., Sinha, S., Baaklini, C.S., Dong, Y., Ho, M., Thorburn, K., *et al.* (2020). Microglia response following acute demyelination is heterogeneous and limits infiltrating macrophage dispersion. *Sci Adv* 6, eaay6324.
- Savage, J.C., Picard, K., González-Ibáñez, F., and Tremblay, M.-È. (2018). A Brief History of Microglial Ultrastructure: Distinctive Features, Phenotypes, and Functions Discovered Over the Past 60 Years by Electron Microscopy. *Frontiers in Immunology* 9.
- St-Pierre, M.-K., Šimončičová, E., Bögi, E., and Tremblay, M.-È. (2020). Shedding Light on the Dark Side of the Microglia. *ASN Neuro* 12, 1759091420925335.
- Viana, M.P., Chen, J., Knijnenburg, T.A., Vasan, R., Yan, C., Arakaki, J.E., Bailey, M., Berry, B., Borensztein, A., Brown, E.M., *et al.* (2023). Integrated intracellular organization and its variations in human iPS cells. *Nature* 613, 345-354.

REVIEWERS' COMMENTS

Reviewer #1 (Remarks to the Author):

This version is more complete than the original submission. The authors have addressed the comments of all reviewers and have significantly improved the manuscript. The ultrastructural characterization is more complete, methodical, and thorough. The authors also made a better effort to link ultrastructural features to ST-defined clusters. Only a few concerns remain.

Introduction:

In contrast, electron microscopy (EM) provides nanometer-resolution view of tissue ultrastructure but lacks molecular readouts.

It should be noted that EM does not lack molecular readout, for instance there is immuno EM but it is low throughput and some antibody staining is not compatible with EM.

At one point the authors refer to cluster 0 (of EM analysis) as reactive. The recommendation stands to avoid the term reactive as this term has been used to describe microglia that are responding to many and different types of stimuli that do not always cause the same type of change in microglia. I do recognize that finding another term is also difficult. My suggestion would be to make an annotation that the word reactive is chosen, for a lack of a better word, but that reactive microglia represent a concept rather than an ultrastructural feature.

Reviewer #3 (Remarks to the Author):

Firstly, I would like to commend your efforts in developing a feature-based quantitative analysis of EM data to characterize and classify single cell based on morphological features. It provides valuable insights into microglia classification.

However, I have one minor point to bring up. I am curious about the feature selection process. Were these features specifically chosen to enhance microglia classification, or can they also be applied to other cell types? Providing some clarity on this point would be beneficial to readers looking to apply your methodology to other types of cells.

REVIEWERS' COMMENTS

Reviewer #1 (Remarks to the Author):

This version is more complete than the original submission. The authors have addressed the comments of all reviewers and have significantly improved the manuscript. The ultrastructural characterization is more complete, methodical, and thorough. The authors also made a better effort to link ultrastructural features to ST-defined clusters. Only a few concerns remain.

We are grateful for the reviewers' positive feedback on our revised manuscript.

Introduction:

In contrast, electron microscopy (EM) provides nanometer-resolution view of tissue ultrastructure but lacks molecular readouts.

It should be noted that EM does not lack molecular readout, for instance there is immuno EM but it is low throughput and some antibody staining is not compatible with EM.

We agree with the reviewer, and we have modified the introduction sentence to reflect this comment as following:

"In contrast, electron microscopy (EM) provides nanometer-resolution view of tissue ultrastructure, but its molecular readouts are limited to techniques like immunogold labeling or correlative light-electron microscopy, which are insufficient for comprehensive characterization of cellular states."

...

"However, techniques that could simultaneously probe cellular ultrastructure and multiplexed molecular profiles in situ are currently unavailable."

Additionally, this thought is further reiterated in discussion of the manuscript (no changes introduced in this version):

*"STcEM tackles this bottleneck and extends the capacity of current EM methods such as correlative light and electron microscopy or immunogold labeling from handful of antibodies to hundreds of molecular targets. This enables the identification of subtle cellular states, and the study of interplay between structural reorganization and gene activity of cells in response to stimuli *in situ*."*

At one point the authors refer to cluster 0 (of EM analysis) as reactive. The recommendation stands to avoid the term reactive as this term has been used to describe microglia that are responding to many and different types of stimuli that do not always cause the same type of change in microglia. I do recognize that finding another term is also difficult. My suggestion would be to make an annotation that the word reactive is chosen, for a lack of a better word, but that reactive microglia represent a concept rather than an ultrastructural feature.

We agree on the complexities around microglia states and nomenclature. We even contributed to a perspective on this important and evolving topic: "Microglia states and nomenclature: A field at its crossroads" (Paolicelli et al., Neuron, 2022). Based on input from

reviewers and our published recommendations, we described microglia by considering multiple layers of complexity, including lipid composition, morphology/ultrastructure, location, and -omics. Despite our efforts, we acknowledged the reviewer's point that our initial version could unintentionally create a misleading association between categories and functions. We adopted the reviewer's suggestion to use the term "reactive," specifically in the context of our demyelination injury model to improve clarity. We have made efforts to define the ultrastructural features upon which our terminology is based when they are first introduced in the manuscript:

“EM annotation revealed multiple microglial subgroups, including normal-appearing microglia (“Homeo-EM”), reactive microglia with high content of lysosomes (“Reactive-lysosomal-EM”), microglia with reactive morphology, high ER and mitochondrial content, but low lysosomal content (“Reactive-nonlysosomal-EM”),

We also describe these features in Table 3 with references to prior work, which we hope offers sufficient clarification for readers, on what we mean by “reactive” morphological classes. We do, however, understand the reviewer's concern about potential misinterpretation of this term. Regarding Cluster 0, this cluster displayed ultrastructural characteristics similar to those we described as "reactive nonlysosomal" (e.g. higher mitochondrial content but without dramatic increase of lysosomes, Figure 4c.) Nonetheless, we have used the term "Cluster 0" in figures and legends to increase clarity. Following the reviewer's suggestion, we have further adjusted our language in the Results section to specify the contextual nature of this cluster:

“Cluster 2 was characterized by high lysosomal content, while Clusters 1 and 0 were less distinctive, and seemed to represent microglia in a more homeostatic and “reactive-nonlysosomal” state, respectively.”

We hope that this revision adequately addresses the reviewer's concern and provides greater clarity for our readers.

Reviewer #3 (Remarks to the Author):

Firstly, I would like to commend your efforts in developing a feature-based quantitative analysis of EM data to characterize and classify single cell based on morphological features. It provides valuable insights into microglia classification.

However, I have one minor point to bring up. I am curious about the feature selection process. Were these features specifically chosen to enhance microglia classification, or can they also be applied to other cell types? Providing some clarity on this point would be beneficial to readers looking to apply your methodology to other types of cells.

We appreciate reviewer's positive feedback on our work. Regarding the feature selection process - we segmented the entire organellar inventory of microglia. This was the most time-consuming step and can be done with any other cell type. There are, however, certain structures that are typical for microglia, at least in the extent of their prevalence. This concerns microglial foam cells which are quite unique in their high lipid droplet content. In contrast, astrocytes would have to be analysed by their fibrillary and glycogen granules besides the general organellar structures (ER, mitochondria etc). As mentioned in Results, for the clustering analysis, we used areas of all segmented structures together with their

calculated ratios (with removal of redundancies). Thus, this type of analysis could be readily adapted to other cell types as long as their informative structures are quantified. We added further clarification in the Methods section:

“Segmentation

For quantitative feature-based classification of ultrastructural microglial morphologies we segmented the entire organellar inventory of examined cells. These segmentations are cell-type independent, only the extent of prevalence of lipid droplets is characteristic to (foamy) microglia. Specifically,

...

“Clustering analysis

PCA was calculated based on all structural features (areas of segmented ultrastructures and their non-redundant ratios) and first 20 PCs were used...”